# Obstructive sleep apnea mediates genetic risk of Diabetes Mellitus in Hispanic and Latino communities
Yana Hrytsenko [1,2,3], Brian W. Spitzer[1,3], Heming Wang [1,2], Suzanne M. Bertisch[1,2], Kent D. Taylor[4], Olga Garcia-Bedoya[5], Alberto R. Ramos[6], Martha L. Daviglus[7], Linda C. Gallo [8], Carmen R. Isasi [9], Jianwen Cai [10], Qibin Qi [9], Carmela Alcántara[11], Susan Redline [1,2,12] & Tamar Sofer [1,2,3,13] ✉

## Abstract

**Background** Obstructive sleep apnea (OSA), and other sleep disorders, are associated with increased risk of developing diabetes mellitus (DM). We examined whether sleep disorders influence the genetic risk of developing diabetes in Hispanic/Latino individuals.
**Methods** We developed Type 2 Diabetes (T2D) polygenic risk score (T2D-PRS) useful in admixed Hispanic/Latino individuals. We estimated the association of the T2D-PRS with cross-sectional ($n = 12,342$) and incident ($n = 6965$) DM in the Hispanic Community Health Study/Study of Latinos (ages 18–76, 50.9% female). We conducted a mediation analysis with T2D-PRS as an exposure, incident DM as an outcome, and OSA as a mediator. Additionally, we performed Mendelian randomization (MR) analysis to assess the causal relationship between T2D and OSA.
**Results** Here, we show that a 1 standard deviation increase in T2D-PRS has DM adjusted odds ratio (OR) = 2.67, 95% CI [2.40; 2.97] and a higher incident DM rate (incident rate ratio (IRR) = 2.02, 95% CI [1.75; 2.33]). In a stratified analysis based on OSA severity categories the associations are stronger in individuals with mild OSA compared to those with moderate to severe OSA. Mediation analysis suggests that OSA mediates the T2D-PRS association with DM. In two-sample MR analysis, T2D has a causal effect on OSA, OR = 1.03, 95% CI [1.01; 1.05], and OSA has a causal effect on T2D, with OR = 2.34, 95% CI [1.59; 3.44].
**Conclusions** These results support a causal association between OSA and DM, with OSA mediating up to 4.7% of the genetic risk for DM. OSA treatment may reduce DM prevalence.

## Plain Language Summary

Sleep disorders such as obstructive sleep apnea play a role in diabetes. Here, we used genetic data to assess this role in over 12,000 Hispanic/Latino individuals. We developed a polygenic risk score summarizing genetic liability to diabetes, and showed that individuals with higher values of this genetic risk score are more likely to develop diabetes. We found that some of the genetic risk of diabetes is mediated via obstructive sleep apnea. Additionally, we implemented a genetic data analysis method called Mendelian randomization, finding that obstructive sleep apnea causally increases risk of diabetes. These results suggest that further studies should consider treating obstructive sleep apnea to potentially reduce diabetes prevalence.

There is compelling evidence linking obstructive sleep apnea (OSA) to Diabetes Mellitus (DM), a group of metabolic diseases marked by elevated blood glucose levels due to defects in insulin secretion and utilization[1]. OSA is a common sleep-related breathing disorder characterized by repeated episodes of upper airway obstruction associated with intermittent hypoxemia and fragmented sleep- mechanisms that are implicated in impaired glucose regulation[2]. The potential pathways linking OSA and DM and evidence for a causal association have been reported previously[3–6]. For example, it was shown that OSA-related intermittent hypoxia and fragmented sleep increase the risk of DM via effects on chronic inflammation,

[1]Department of Medicine, Brigham and Women's Hospital, Boston, MA, USA. [2]Department of Medicine, Harvard Medical School, Boston, MA, USA. [3]Cardio-Vascular Institute (CVI), Beth Israel Deaconess Medical Center, Boston, MA, USA. [4]The Institute for Translational Genomics and Population Sciences, Department of Pediatrics, The Lundquist Institute for Biomedical Innovation at Harbor-UCLA Medical Center, Torrance, CA, USA. [5]Division of Academic Internal Medicine and Geriatrics, College of Medicine, University of Illinois Chicago, Chicago, IL, USA. [6]Department of Neurology, University of Miami Miller School of Medicine, Miami, FL, USA. [7]Institute for Minority Health Research, Department of Medicine, College of Medicine University of Illinois Chicago, Chicago, IL, USA. [8]Department of Psychology, San Diego State University, San Diego, CA, USA. [9]Department of Epidemiology & Population Health, Albert Einstein College of Medicine, New York, NY, USA. [10]Department of Biostatistics, University of North Carolina at Chapel Hill, Chapel Hill, NC, USA. [11]School of Social Work, Columbia University, New York, NY, USA. [12]Department of Epidemiology, Harvard T.H. Chan School of Public Health, Boston, MA, USA. [13]Department of Biostatistics, Harvard T.H. Chan School of Public Health, Boston, MA, USA. ✉e-mail: tsofer@bidmc.harvard.edu

oxidative stress and metabolic imbalance that adversely affect pancreatic function, hormonal regulation, and insulin resistance[7–10]. In the Hispanic Community Health Study/Study of Latinos (HCHS/SOL), prospective analyses demonstrated that OSA was associated with an approximately 30% increased incidence of DM[11]. In addition to OSA, other metrics of poor sleep, including chronic partial sleep loss, have been related to DM risk[12]. Meta-analyses showed that quantity and quality of sleep, including short and long sleep durations, increase the risk of development of type 2 diabetes[13,14]. A study in HCHS/SOL found that those with short sleep and insomnia, and long sleep without insomnia had elevated odds of diabetes prevalence[15]. In addition, several cohort studies implicate T2D -particularly insulin resistance- as a risk factor for OSA, due to effects of insulin on fat distribution and pharyngeal collapsibility[16,17]. DM is one of the primary risk factors for cardiovascular diseases, which in turn, is the leading cause of mortality in Hispanics/Latinos individuals[18] and thus, elucidation of the complex interplay between sleep and DM is particularly important in Hispanic/Latino populations.

Recent analysis in HCHS/SOL utilized genetic techniques to study the association of OSA with a range of phenotypes. The study showed that a polygenic risk score for OSA was associated with glycemic traits[19]. Additionally, the same study applied two-sample Mendelian randomization (MR) analysis using GWAS summary statistics, and suggested a causal effect of DM-related glycemic traits on OSA. Indeed, several large cohort studies have reported observational evidence of a bidirectional association between OSA and diabetes[16]. However, this MR analysis found no statistically significant association of OSA on DM-related traits. This null finding suffers from limitations: the MR used OSA GWAS summary statistics from a population of Finnish Europeans, and included only a small number of genetic loci. In addition, the association between OSA and diabetes risk may involve more complex mechanisms such as effect modification. For example, OSA may interact with genetic risk factors, affecting the likelihood of developing diabetes.

We hypothesized that OSA interacts with genetic risk for DM to increase DM risk. Such interactions are consistent with recent data demonstrating interactions between sleep duration and genetic loci in associations for blood pressure and lipid levels; specifically, multiple genome-wide gene-environment studies identified interactions of genomic loci with interaction with either short or long sleep duration in relation to blood pressure and lipid measures[20,21]. It is similarly possible that OSA increases the risk of DM also via modification of genetic effects. Polygenic risk scores (PRSs) are increasingly used to summarize the genetic liability of a disease[22–28]. Given the strong association between OSA and DM, it is important to study the risk conferred by PRS for DM within the context of the "biological environment" of OSA, where we assume that individuals with OSA have potentially different tissue physiological functions compared to individuals without OSA due to hypoxia, for example.

Hispanics and Latinos are of admixed origin, with three predominant ancestral populations of European (EUR), African (AFR), and Amerindian (AMR) ancestries[29]. Thus, the genomes among groups of people of Hispanic and Latino background are a mosaic of genomic intervals each inherited from AMR, EUR, and AFR ancestries. Until recently, most studies of genetic susceptibility to DM have been performed in cohorts of European or Asian ancestry[30,31], which may not fully represent the ancestral mosaic in Hispanic, admixed individuals. Recently, several large studies of genetic susceptibility to DM via multi-ancestry meta-analysis have identified hundreds of loci associated with type 2 diabetes[32,33] and laid the foundation for developing PRS that are useful across diverse populations, including admixed Hispanics/Latinos. Thanks to the availability of ancestry-specific GWAS, here we develop new polygenic risk scores (PRSs) for DM that are useful for admixed individuals such as Hispanics/Latinos, and study whether their association with DM is modified by OSA or, in secondary analysis, other poor sleep phenotypes.

The primary PRS that we developed is highly associated with DM, in both prevalence (odds ratio = 2.67) and incident (incident rate ratio = 2.02) associations. The PRS is associated with OSA, but not with other sleep

phenotypes examined, supporting that OSA is the strongest sleep-related heritable risk factors to DM, of those examined, in the Hispanic/Latino population. Mediation analysis suggests that mild-to-severe OSA mediates up to 4.7% of the genetic risk of DM, and Mendelian randomization analysis supports that OSA is a causal risk factor for DM.

## Methods

We used summary statistics from published GWAS and individual-level data from the Mass General Brigham (MGB) Biobank to develop T2D-PRS in different ways. We used genetic data to study the associations between OSA (and, in secondary analysis, other sleep measures) and DM using data from HCHS/SOL. We applied a number of approaches for analysis, including PRS, MR, and mediation analysis. Because both OSA and T2D are heavily impacted by obesity, we further considered summary statistics for PRS analyses based on BMI-adjusted analyses. We applied bidirectional MR and multivariate MR (MVMR) to assess the causal relationship between T2DM and OSA, with and without adjusting for BMI. Tables 1 and 2 summarizes the genetic data and analyses performed, and Supplementary Data 2 provides the SNP-level data for the MVMR analysis.

### Development of polygenic risk scores for T2D

For the primary T2D-PRS developed, we used summary statistics from two large GWAS efforts of T2D: the DIAGRAM consortium[32] and the Million Veteran Program (MVP)[33], where T2D GWAS were not adjusted for BMI. Both sets of summary statistics were based on individuals from multiple populations and genetic ancestries. The DIAGRAM consortium provided summary statistics from GWAS meta-analysis in East Asian (EAS), European (EUR), and South Asian (SAS) individuals. MVP provided summary statistics from analysis of White (EUR), Black (AFR), and Hispanic (AMR) HARE (harmonized ancestry and race/ethnicity) groups. Details are provided in Supplementary Table 1. We first meta-analyzed the DIAGRAM-European and MVP-White summary statistics using inverse-variance weighted meta-analysis implemented in GWAMA[34]. Next, we used PRS-CSx[35] (global shrinkage parameter $\phi$ was learnt from the data, other parameters left at default) to develop ancestry-specific PRS for EUR, AFR, EAS, SAS, and AMR groups (now using population descriptors provided by the Linkage Disequilibrium reference panels implemented by PRS-CSx; specifically, we used those based on the 1000 Genomes reference panels), focusing on HapMap SNPs (as available in the PRS-CSx provided reference panel data)[36]. This resulted in a list of variants and weights for each of EUR, AFR, EAS, SAS and AMR T2D-PRSs. Because of the specific admixture patterns in HCHS/SOL individuals, we only moved forward with EUR, AFR, and AMR ancestry-specific T2D-PRSs in data analysis, as described latter.

In secondary analysis we developed T2D-PRS based on European ancestry individuals only using the DIAGRAM consortium summary statistics from a BMI-adjusted GWAS[30]. Other summary statistics from BMI-adjusted analyses are not available. Thus, we used PRS-CS to develop PRS weights from the T2D BMI-adjusted GWAS employing LD-reference panel derived from European UKBB individuals. We used the maximum Neff (total reported effective sample size) reported for this data set (157,390) as the sample size, and allowed PRS-CS to learn the global shrinkage parameter $\phi$ from the data. All other parameters were left at their default values. The posterior effect size estimates generated by PRS-CS are the PRS weights. This PRS is referred to as BMIadjT2D-PRS.

### The Hispanic Community Health Study/Study of Latinos

The HCHS/SOL is a population-based cohort study of Hispanic/Latino adults in the United States. Individuals were recruited to the study via a multi-stage sampling design, as previously described[37,38]. The study enrolled 16,415 adult participants (18- to 74-year-old at baseline) from four geographic areas: Bronx, NY, Chicago, IL, Miami, FL, and San Diego, CA, with enrollment between 2008-2011 (visit 1). Individuals self-identified with Hispanic/Latino backgrounds including Cuban, Central American, Dominican, Mexican, Puerto Rican, and South American. During the

**Table 1 | Outline of the PRS analyses performed in this study**

| **PRS associations with DM at baseline** | | | | |
|---|---|---|---|---|
| **Analysis** | **GWAS** | **PRS type** | **PRS method** | **Stratification categories** |
| T2D-PRS association with DM at baseline: newly developed PRSs. | BMI-unadjusted multi-ancestry T2D GWASs: DIAGRAM, MVP BMI-adjusted European T2D GWAS: DIAGRAM | mgbPRSsum gapPRSsum PRSsum BMIadjPRS | PRS-CSx PRS-CS | Primary: Overall dataset, OSA severity categories Secondary: sleep phenotypes categories, Hispanic/Latino background |
| T2D-PRS association with DM at baseline: compare to existing PRSs | BMI-unadjusted multi-ancestry T2D GWASs | PGS003867_PRS PGS002308_PRS | PRS-CSx PRS-CS | Overall dataset |
| OSA-PRS association with DM at baseline | BMI-adjusted and BMI-unadjusted multi-ancestry OSA GWAS: MVP | Genome-wide SNPs (LDPred2) | LDpred2 | Overall dataset |
| **PRS associations with incident DM** | | | | |
| T2D-PRS association with incident DM: newly developed PRSs. | BMI-unadjusted multi-ancestry T2D GWASs: DIAGRAM, MVP BMI-adjusted European T2D GWAS: DIAGRAM | mgbPRSsum gapPRSsum PRSsum BMIadjPRS | PRS-CSx PRS-CS | Primary: Overall dataset, OSA severity categories Secondary: sleep phenotypes categories, Hispanic/Latino background |
| T2D-PRS association with incident DM: comparison to existing PRSs | BMI-unadjusted multi-ancestry T2D GWASs | PGS003867_PRS PGS002308_PRS | PRS-CSx PRS-CS | Overall dataset |
| **PRS associations with poor sleep phenotypes at baseline** | | | | |
| T2D-PRS association with poor sleep phenotypes | BMI-unadjusted multi-ancestry T2D GWASs: DIAGRAM, MVP BMI-adjusted European T2D GWAS: DIAGRAM | mgbPRSsum gapPRSsum PRSsum BMIadjPRS | PRS-CSx PRS-CS | Primary: OSA severity categories, sleep phenotypes categories Secondary: adjusting for WHR; including additional potential confounders - medication, background, physical activity and socioeconomic status |
| OSA-PRS association with OSA | BMI-unadjusted and BMI-adjusted multi-ancestry OSA GWAS: MVP | Genome-wide SNPs (LDPred2) | LDpred2 | OSA severity categories |
| **T2D-PRS interaction analysis** | | | | |
| Interaction between T2D-PRSs and OSA severity categories | BMI-unadjusted multi-ancestry T2D GWASs: DIAGRAM, MVP | mgbPRSsum gapPRSsum PRSsum | PRS-CSx | OSA severity categories |
| **Mediation analysis** | | | | |
| Mediation analysis: T2D-PRS → OSA → incident DM | BMI-unadjusted and BMI-adjusted multi-ancestry T2D GWASs: DIAGRAM, MVP | mgbPRSsum gapPRSsum PRSsum | PRS-CSx | OSA, REI |

mgbPRSsum, gapPRSsum, and PRSsum were constructed as weighted sums of the standardized ancestry-specific PRSs for each individual. In mgbPRSsum the weights were the estimated coefficients of the three PRSs in a logistic regression of DM over the PRSs, age, sex, and 10 PCs in the MGB Biobank. In gapPRSsum the weights were individual-specific, using the individual's estimated proportions of global genetic ancestry. In PRSsum the weights were all equal to 1. BMIadjPRS was developed based on the BMI-adjusted T2D GWAS.
PGS003867_PRS and PGS002308_PRS were previously developed (variants and weights obtained from the PGS catalog).
OSA-PRS was constructed based on previously-reported variants and weights. Bold font represents column names and analysis categories.
*BMI* body mass index, *DIAGRAM* diabetes genetics replication and meta-analysis consortium, *DM* diabetes mellitus, *GIANT* the Genetic investigation of anthropometric traits consortium, *GWAS* genome-wide association study, *LDpred2* linkage disequilibrium aware PRS model, *MVP* million veteran program, *OSA* obstructive sleep apnea, *PRS* polygenic risk score, *PRS-CS* PRS-continuous shrinkage, *PRS-CSx* cross-population continuous shrinkage PRS, *REI* respiratory even index, *SNP* single nucleotide polymorphism, *T2D* type 2 diabetes.

baseline exam (visit 1), individuals responded to various questionnaires, including sleep-related, and health measures including anthropometry, scanned medications, and fasting blood samples, were collected. HCHS/SOL participants were invited to participate in a second visit (visit 2; N = 11,623), which took place from 2014–2017, on average 6 years following visit 1. HCHS/SOL analyses typically used weights, generated by the HCHS/SOL Coordinating Center, in order to generate estimates that are applicable to the HCHS/SOL target population. Separate weights are available for the baseline visit 1 and for visit 2, accounting for loss to follow up. Statistical analyses used weights depending on the data used (visit 1, visit 2), and utilized all individuals with appropriate data measured and without missing values.

### Sleep measures
At visit 1, participants underwent home sleep testing with an ARES Unicorder 5.2 (B-Alert, Carlsbad, CA) device within a week of their exam. The device measured nasal airflow, heart rate, snoring, body position, and oxyhemoglobin saturation. Based on the device measurements, the respiratory event index (REI) was calculated as the number of respiratory events (defined as at least 50% reduction in airflow with at least 3% desaturation for 10 s or more) per estimated sleep hour. OSA severity was defined based on the REI, with mild OSA defined as $15 \geq REI \geq 5$, and moderate-to-severe OSA defined as $REI \geq 15$. $REI < 5$ was considered no OSA. More information on the sleep study is provided in Redline et al.[39].

Other sleep phenotypes were self-reported and included insomnia, defined by the Women's Health Initiative Insomnia Rating Scale[40] WHIIRS $\geq 10$, short sleep, defined by the average sleep duration in hours $\leq 6$, long sleep, defined by sleep duration > 9, excessive daytime sleepiness (EDS), defined by the Epworth Sleepiness Scale[41] ESS > 10. Sleep duration was assessed through questions regarding typical wake and bedtimes on weekdays and weekends: "What time do you usually go to bed (on weekdays/weekends)?" and "What time do you usually wake up (on weekdays/weekends)?" The average sleep duration was then calculated as a weighted average, with weekday sleep duration weighted at 5/7 and weekend sleep duration at 2/7.

**Table 2 | Outline of the Mendelian randomization analyses performed in this study**

| Analysis | GWAS | IVs | SNPs *p*-value threshold | MR-method |
|---|---|---|---|---|
| Mendelian Randomization: T2D → OSA | BMI-unadjusted and BMI-adjusted European T2D and OSA GWASs: DIAGRAM, MVP-White | SNPs significantly associated with T2D | $5 \times 10^{-8}$<br>$10^{-7}$<br>$10^{-5}$ | Primary: IVW, MR-RAPS, MR-PRESSO Secondary: MR-Egger, weighted median, simple mode, weighted mode, MR-RAPS. |
| Mendelian Randomization: OSA → T2D | BMI-unadjusted and BMI-adjusted European T2D and OSA GWASs: DIAGRAM, MVP-White | SNPs significantly associated with OSA | $5 \times 10^{-8}$<br>$10^{-7}$<br>$10^{-5}$ | Primary: IVW, MR-RAPS, MR-PRESSO Secondary: MR-Egger, weighted median, simple mode, weighted mode, MR-RAPS. |
| Multivariable Mendelian Randomization: T2D → OSA | BMI-unadjusted European T2D, BMI and OSA GWASs: DIAGRAM, GIANT, MVP-White | SNPs significantly associated with T2D or BMI | $5 \times 10^{-8}$<br>$10^{-7}$<br>$10^{-5}$ | Primary: IVW |
| Multivariable Mendelian Randomization: OSA → T2D | BMI-unadjusted European OSA, BMI and T2D GWASs: DIAGRAM, GIANT, MVP-White | SNPs significantly associated with OSA or BMI | $5 \times 10^{-8}$<br>$10^{-7}$<br>$10^{-5}$ | Primary: IVW |

*BMI* body mass index, *DIAGRAM* diabetes genetics replication and meta-analysis consortium, *GIANT* the Genetic investigation of anthropometric traits consortium, *IV* instrumental variable, *IVW* inverse-variance weighted MR estimator, *MR* Mendelian randomization, *MR-Egger* MR using Egger regression, *MR-RAPS* Mendelian randomization using the robust adjusted profile score, *MR-PRESSO* Mendelian randomization pleiotropy residual sum and outlier, *MVP* million veteran program, *OSA* obstructive sleep apnea, *SNP* single nucleotide polymorphism, *T2D* type 2 diabetes.

## DM and incident DM outcomes

Diabetes status was ascertained based on American Diabetes Association (ADA) definition or scanned medication (at the baseline exam) or self-reported diabetes medication use (at the second exam, as scanned medications were not available). ADA criteria rely on laboratory tests, either fasting time >8 and fasting glucose ≥126 mg/dL, or fasting time ≤8 and fasting glucose ≥ 200 mg/dL, or post-OGTT glucose ≥ 200 mg/dL, or A1C ≥6.5%. Hyperglycemia (pre-diabetic status) was defined by the ADA guidelines based on laboratory tests of fasting time >8 and fasting glucose in range 100 to 125 mg/dL, or post-OGTT glucose in range 140 to 199 mg/dL, or 5.7%≤ A1C <6.5%. Individuals had incident DM if they did not have DM at visit 1 and had DM at visit 2. Descriptions of the scanned anti-diabetic medications and self-reported antidiabetic medication use are provided in Supplementary Note 1. Diabetes status was considered missing if fasting glucose, post-OGTT, HbA1c, and information on anti-diabetic medications were all unavailable. Participants without glucose lab data and no record of anti-diabetic medication use were assumed to have normal glucose regulation.

## Genotyping, imputation, and PRS construction

Consented HCHS/SOL individuals were genotyped using the Illumina Custom 15041502 B3 array as previously described[29,42]. Quality control was performed, including checks that biological sex matched reported gender. As described in Conomos et al.[29], genetic principal components (PCs) and kinship matrix, tabulating genetic relationship between individuals, were computed using PC-AiR and PC-Relate, implemented in the GENESIS R package[43,44]. Proportions of continental ancestry were estimated as previously reported via model-based analysis using the ADMIXTURE software[45] under the assumption of four ancestral populations (West African, European, Amerindian and East Asian). Consequently, a small number of individuals with East Asian ancestry were removed, and the analysis was repeated with three ancestral populations (excluding East Asian). Genome-wide imputation via the Michigan imputation server[46] was conducted using the TOPMed 2.0 imputation panel. Only variants with imputation quality R2 ≥ 0.8, minor allele frequency ≥0.01, and not strand-ambiguous, were used in PRS construction. All PRSs were constructed in HCHS/SOL from lists of variants, alleles, and weights, using the PRSice software[47], without any clumping and thresholding.

For primary analysis, we constructed in HCHS/SOL the three ancestry-specific PRSs developed by PRS-CSx as described earlier (EUR, AFR, and AMR). The three PRSs were standardized to have mean 0 and variance 1 in the dataset. Next, we created three PRSs for each individual, summing the three ancestry-specific PRSs in different ways: (1) gapPRSsum: a weighted

sum of EUR, AFR, and AMR-specific standardized PRSs, weighted by an admixed individual's estimated proportions of global genetic ancestry; (2) PRSsum: an unweighted sum, where the three standardized ancestry-specific PRSs were summed without any weights; and (3) mgbPRSsum: a weighted sum of the standardized ancestry-specific PRSs for each individual, where weights were computed as the estimated coefficients of the three PRSs in a logistic regression of DM over the PRSs, age, sex, and 10 genetic principal components (PCs), in the Mass General Brigham (MGB) Biobank (described below). After summing the ancestry-specific PRS, we again standardized each resulting PRS measure. These standardized PRSs were subsequently used and were not transformed again.

For performance comparison, we constructed two additional T2D-PRSs from the list of effect variants and their respective effect sizes based on multi-ancestry T2D GWASs reported by Shim et al.[48] and Ge et al.[49]. The list of effect variants and their effect sizes were downloaded from the PGS catalog[50]. From here on, we refer to them as PGS003867_PRS and PGS002308_PRS, using the PRS identifiers from the PGS catalog. These PRSs were also standardized.

We compared each model's prediction performance based on including only standard covariates (age, sex, BMI, study center, and genetic PCs) with models that included T2D-PRSs in addition to the above covariates. We computed the area under the receiver operating characteristic curve (AUC) and its 95% confidence interval (CI) for each model using repeated random train-test splits, to estimate model performance on an independent dataset (rather than on a training dataset, which may lead to optimism due to overfitting). Specifically, we randomly split the data into training (90%) and testing (10%) sets, trained the model on the training data, and evaluated its performance on the testing data using the *predict* function in R. This process was repeated 500 times, and the AUC was computed using the auc function from the Metrics R library (version 0.1.4). The mean AUC was reported, and the 95% CI was defined using the 2.5 and 97.5 percentiles of the 500 AUC values.

## The Mass General Brigham Biobank

Samples, genomic data, and health information were obtained from the Mass General Brigham (MGB) Biobank, a biorepository of consented patient samples at Mass General Brigham. MGB researchers are eligible for accessing data from the biobank via a web application. All Biobank subjects have provided their consent to join the MGB Biobank, which includes agreeing to provide a blood sample linked to the electronic medical record. Subjects also agree to be recontacted by the MGB Biobank staff as needed. The MGB Biobank was approved by the MGB Institutional Review Board.

### Genotyping, imputation, and PRS construction at MGB biobank

DNA samples are processed from whole blood that was collected as a dedicated research draw or as a clinical discard. Dedicated research samples are aimed to be processed within four hours of collection. Clinical discards are processed 24+ hours after collection. Whole blood is spun to buffy coat with a centrifuge and the buffy coat is stored in a freezer up to several months. The buffy coat is then extracted to DNA. The DNA is then placed in an ultralow freezer ($-80\,°C$). Each DNA aliquot contains a minimum of 2 μg of DNA. The concentration varies.

Samples have been genotyped using three versions of the biobank SNP array offered by Illumina that is designed to capture the diversity of genetic backgrounds across the globe. The first batch of data was generated on the Multi-Ethnic Genotyping Array (MEGA) array, the first release of this SNP array. The second, third, and fourth batches were generated on the Expanded Multi-Ethnic Genotyping Array (MEGA Ex) array. All remaining data were generated on the Multi-Ethnic Global (MEG) BeadChip.

Prior to performing imputation, files were converted to VCF format, separated by chromosomes. When multiple probes measured the same genotypes, they were checked for concordance and were set to a missing value if the genotypes did not match. Files were uploaded to the Michigan Imputation Server, and Genotypes were imputed using TOPMed reference panel. Genomic coordinates are provided in GRCh38.

Quality control was performed using PLINK (v2.0). SNPs with low-quality imputation ($r < 0.5$), with missing call rates $> 0.1$, HWE $p$-value less than $1 \times 10^{-6}$ and MAF $< 1\%$ were filtered out. Principal components (PC) were computed using PLINK: pruning of genotype data was done using a window size of 1000 variants, sliding across the genome with a step size of 250 variants at a time, filtering out any SNPs with LD $R^2 > 0.1$. Computation of loadings for the first 10 PCs was done using unrelated individuals (3rd degree, identified using PLINK).

PRSs were calculated with PRSice (with no clumping and thresholding). All of the parameters were left at default values.

### Diabetes status at the MGB Biobank based on Curated Disease Populations

We used the diabetes outcome from the "curated disease populations" provided by the MGB Biobank team. These phenotypes were developed by the Biobank Portal team using both structured and unstructured electronic medical record (EMR) data and clinical, computational and statistical methods. Natural Language Processing (NLP) was used to extract data from narrative text. Chart reviews by disease experts helped identify features and variables associated with particular phenotypes and were also used to validate results of the algorithms. The process produced robust phenotype algorithms that were evaluated using metrics such as sensitivity, the proportion of true positives correctly identified as such, and positive predictive value (PPV), the proportion of individuals classified as cases by the algorithm [1]. The high throughput phenotyping algorithm is as follows:

1. Create an initial phenotype definition using ICD-9 diagnosis codes.
2. Broaden the definition by determining the most up-to-date features (comorbidities, symptoms, medications) that create a more accurate profile of the phenotype when combined with ICD-9 codes. Features are extracted from online medical literature and knowledge bases via an Automated Feature Extraction Protocol (AFEP).
3. Narrow and refine the definition by determining the features that occur most often in the Biobank data. Extract, code, and rank features contained in clinical narratives with Natural Language Processing (NLP).
4. Create a gold-standard patient set for training the method. Query coded EMR data for the set of patients having at least one ICD-9 code for the phenotype. Apply a statistical sampling algorithm to select a random subset of those patients for full chart review. A clinical expert performs a full chart review to classify the patients as positive or negative for the phenotype.
5. Train a statistical model that incorporates all features in the definition to predict the presence or absence of the phenotype against the gold-standard patient set.
6. Apply the trained model to the entire Biobank Population.

There were and 40,209 genetically unrelated individuals. We then restricted the dataset to the individuals who are not deceased of 18 years or older reducing the data to 36,423 individuals. Of these, 2934 (8%) had T2D according to the MAP algorithm. BMI values were taken to be the average BMI across all values available for an individual.

### Association analysis of T2D-PRS with DM and incident DM in HCHS/SOL

We first verified that the T2D-PRSs were associated with DM by performing the association test between T2D-PRSs with DM at visit 1. We used survey logistic regression with DM as the outcome and adjusted for age, sex, BMI, field center and the first 5 principal components of genetic data to account for potential population stratification. This analysis used survey weights computed based on visit 1 participation. We next estimated the association of the T2D-PRSs with incident DM using analysis restricted to participants who did not have DM at visit 1, in the combined dataset and stratified by Hispanic/Latino background and by sleep phenotypes. Here, we performed survey Poisson regression accounting for visit 2 survey weights and using the time between the baseline clinic visit and visit 2 as an offset. Otherwise, covariates were the same as described for the baseline DM status association analysis.

Association analyses were performed among all available individuals and restricted to strata of OSA categories. In secondary analysis, we performed association analyses between T2D-PRS and DM and incident DM, stratified by other categories defined by long and short sleep durations, insomnia, and sleepiness, and stratified (separately) by self-reported Hispanic/Latino background. To reduce the number of displayed associations, secondary analyses focused on mgbPRSsum. We chose this PRS because global ancestry proportions were available only for a genetically unrelated set of individuals, reducing power, and because unweighted sums have been performing less well than the weighted sums in previous work[24].

When it appeared that T2D-PRS association differed by sleep stratum, we also performed interaction analysis, and estimated the multiplicative interaction effect between T2D-PRSs and the relevant sleep strata.

### Association analysis of T2D-PRS with sleep phenotypes in HCHS/SOL

We estimated the associations of the derived multi-ancestry T2D-PRSs with poor sleep phenotypes (OSA: primary, other phenotypes: secondary) using visit 1 data. To test the association of the T2D-PRSs with sleep phenotypes, we used survey logistic regression with baseline survey weights. We set sleep phenotype as the outcome and adjusted for age, sex, BMI, field center and the first 5 genetic PCs. We also performed secondary analyses of the association between T2D-PRSs with OSA (we considered both moderate-to-severe versus mild or no OSA, and mild-to-severe versus no OSA). In one sensitivity analysis we replaced BMI with waist-to-hip ratio (WHR) as a covariate, because WHR, as a measure of central obesity, may more accurately capture the common cause of T2D and OSA. In another sensitivity analysis we included additional potential confounders as covariates in the models: use of statins, measures of socioeconomic status (education level: no high school diploma or GED, at most a high school diploma or GED, greater than high school (or GED) education; household income: <$30,000 or ≥$30,000), physical activity (total physical activity and total vigorous physical activity in a week), and self-reported Hispanic/Latino backgrounds. Each of these covariates was included in a separate regression model, and in addition to the baseline covariates that have been used throughout.

### Mediation analysis of T2D-PRS as the exposure, OSA as a mediator, and incident DM as an outcome in HCHS/SOL

We performed a mediation analysis using individuals without DM at visit 1 who participated in visit 2. Here, we set mgbPRSsum as the exposure and mild-to-severe (versus no) OSA as a mediator of the mgbPRSsum effect on DM. In secondary analysis, we also used BMIadjT2D-PRS as an exposure,

and moderate-to-severe OSA versus no and mild OSA, as well as the continuously measured respiratory event index (REI, log transformed to achieve approximate normality) as mediators (each separately). We used survey logistic regression (linear, for REI) to fit the mediator-exposure association model, and survey Poisson regression to fit the outcome-mediator regression (here, we used the time between visits 1 and 2 to adjust for differences in follow-up duration). All models were adjusted for age, sex, BMI, and 5 genetic PCs and used visit 2 sampling weights. We used the *mediation* R package version 4.5.0 to fit the causal mediation analysis model. Because the T2D-PRSs are continuous, we computed percent mediated effect for increasing the value of the PRS from a "control" to a "treatment" value (*control.value* and *treat.value* in the code). For a given PRS, we computed the values of quantiles (0, 0.25, 0.5, 0.75, and 1) and used these. Thus, we estimated the proportion of the T2D-PRS effect on DM risk that is mediated through increased risk of mild-to-severe OSA, comparing individuals whose T2D-PRS values correspond to, as an example, the 0.25 quantile (control value) versus the 0.75 quantile (treatment value) of the PRS distribution.

### Two-sample Mendelian randomization analysis of T2D and OSA

Using summary statistics from published GWAS (not HCHS/SOL data), we performed bi-directional Mendelian randomization (MR) analysis using the *TwoSampleMR* R package version 0.5.11 to estimate the causal effect between T2D and OSA. We used BMI-adjusted and BMI-unadjusted European ancestry GWASs for both T2D and OSA[30,32,51]. For each exposure-outcome and BMI adjustment combinations, we selected a set of instrumental variables (IVs) from the exposure GWAS by: (1) taking the intersection of SNPs between the exposure and outcome GWASs; (2) filtering the resulting SNPs by their *p*-value in the exposure GWAS ($p < 5 \times 10^{-8}$ and, secondary, $10^{-7}$); (3) performing clumping of the SNPs that passed the *p*-value threshold using the 1000 genomes European reference panel, including only bi-allelic SNPs with MAF > 0.01, and setting clumping window as $kb = 10,000$ and $r2 = 0.001$. Next, we harmonized the exposure and outcome data for these SNPs to ensure that the effect of a SNP on an outcome and exposure is relative to the same allele. We performed MR analysis based on these harmonized data. The primary method was the inverse variance weighted (IVW) random effects meta-analysis. We also used MR-PRESSO[52] because it allows for evaluation of horizontal pleiotropy in multi-instrument MR (we set *NbDistribution* = 10,000 and *SignifThreshold* = 0.05), and MR-RAPS because it is useful when instruments are weak (e.g., $p < 10^{-7}$, rather than $5 \times 10^{-8}$).

### Multivariable Mendelian randomization analysis of T2D and OSA

Using BMI-unadjusted summary statistics from published T2D and OSA GWASs described above and also GWAS of BMI[53], we conducted bidirectional multivariable MR (MVMR) analyses using the *TwoSampleMR* R package version 0.5.11 to estimate the causal relationship between T2D and OSA, adjusting for BMI as an additional exposure. For each of the two exposures and the outcome variables we selected a set of IVs from the exposure GWASs by: (1) taking the intersection of SNPs between the exposures and outcome GWASs; (2) filtering the resulting SNPs by their *p*-value in the two exposure GWASs; (3) taking the union of SNPs between the two exposure GWASs; (4) performing clumping of the SNPs that passed the *p*-value threshold as described above and harmonizing the two exposures and the outcome data for these SNPs to ensure that effect estimates corresponded to the same effect allele across all traits. We performed MVMR analysis based on the harmonized data. From these analyses, we report the estimated causal associations of OSA on T2D adjusted for BMI, and of T2D on OSA, adjusted for BMI. We do not report the estimates of BMI because it is wrong to estimate the effect of BMI on OSA adjusting to T2D (a collider), and similarly for BMI on T2D adjusting for OSA.

### Association analysis of OSA PRSs with T2D in HCHS/SOL

Using BMI-adjusted and BMI-unadjusted multi-ancestry Million Veteran Program[51] OSA GWAS we constructed OSA-PRS (previously developed using LDPred2[54]). To validate the two OSA-PRSs we first estimated the

association of the BMIadjOSA-PRS and BMIunadjOSA-PRS with OSA. Next, to further investigate the potential role of OSA in development of DM we estimated the association of BMIadjOSA-PRS and BMIunadjOSA-PRS with baseline DM. To test the association of the BMIadjOSA-PRS and BMIunadjOSA-PRS with baseline DM, we used survey logistic regression with baseline survey weights. In the OSA-PRSs validation step, we set OSA (mild-to-severe OSA and moderate-to-severe OSA) as the outcome and adjusted for covariates including, age, sex, BMI, field center and 5 genetic PCs. Similarly, we then set baseline DM as the outcome and adjusted the model for age, sex, BMI, field center and the first 5 genetic PCs.

### Statistics and reproducibility

Association analyses were performed in using survey methods to account for the HCHS/SOL study design and produce estimates that are applicable to the HCHS/SOL target population. Associations with a prevalent binary outcome used survey logistic regression, and with incident outcomes used survey Poisson regression, with time between baseline and follow-up visit used as an offset. Association tests used the 1 degree of freedom Wald test statistics, and 95% confidence intervals are reported for the effect estimates. Analyses are reproducible via publicly available code deposited on the linked GitHub repository[55]. No replicates were used as this study is of observational human data.

### Ethics statement

The HCHS/SOL was approved by the institutional review boards (IRBs) at each field center, where all participants gave written informed consent, and by the Non-Biomedical IRB at the University of North Carolina at Chapel Hill, to the HCHS/SOL Data Coordinating Center. All IRBs approving the HCHS/SOL study are: Non-Biomedical IRB at the University of North Carolina at Chapel Hill. Chapel Hill, NC; Einstein IRB at the Albert Einstein College of Medicine of Yeshiva University. Bronx, NY; IRB at Office for the Protection of Research Subjects (OPRS), University of Illinois at Chicago. Chicago, IL; Human Subject Research Office, University of Miami. Miami, FL; Institutional Review Board of San Diego State University, San Diego, CA. All methods and analyses of HCHS/ SOL participants' materials and data were carried out in accordance with human subject research guidelines and regulations. This work was approved by the Mass General Brigham IRB and by the Beth Israel Deaconess Medical Center Committee on Clinical Investigations.

### Reporting summary

Further information on research design is available in the Nature Portfolio Reporting Summary linked to this article.

## Results
### HCHS/SOL participant characteristics

Table 3 characterizes the HCHS/SOL target population, overall and stratified by OSA severity categories (no OSA, mild OSA and moderate to severe OSA). Of the HCHS/SOL target population, 50.9% were females and the mean age and BMI were 41.51 (SD = 15.05) and 29.4 (SD = 6.13), respectively. At visit 1, 48.4% of the target population was normoglycemic, 36.6% was hyperglycemic and 15% met criteria for diabetes. There were only 17 people with anti-diabetes classified based on medication data, who were not classified as diabetics by ADA guidelines. Focusing on the population who did not have DM (normal or hyperglycemic) at visit 1 and participated in visit 2, 8.08% of the HCHS/SOL target population had incident DM at visit 2 (Supplementary Table 2). Characteristics of the HCHS/SOL target population stratified by other sleep phenotype categories are provided in Supplementary Table 3.

### Newly-developed T2D-PRSs are associated with DM and incident DM in the HCHS/SOL

Here, we developed three types of multi-ancestry T2D-PRSs, including PRS constructed as unweighted sum, weighted sum with weights being individual-level ancestral proportions, and weighted sum with weights estimated

**Table 3 | Characteristics of the HCHS/SOL target population at baseline overall and stratified by OSA severity categories**

| Characteristic | All | No OSA | Mild OSA | Moderate to severe OSA |
|---|---|---|---|---|
| **N** | 12,342 | 7563 | 2122 | 1270 |
| **Gender N (%)** | | | | |
| Female | 7244 (50.9) | 4816 (55.4) | 1117 (42.4) | 508 (31.9) |
| Male | 5098 (49.1) | 2747 (44.6) | 1005 (57.6) | 762 (68.1) |
| **Age** | | | | |
| Mean (SD) | 41.51 (15.05) | 37.87 (14.03) | 50.46 (12.83) | 52.62 (12.91) |
| **BMI** | | | | |
| Mean (SD) | 29.40 (6.13) | 28.30 (5.66) | 31.47 (6.05) | 33.71 (6.25) |
| **DM status at baseline N (%)** | | | | |
| Normoglycemic | 5090 (48.4) | 3742 (56.8) | 544 (28.8) | 216 (18.3) |
| Hyperglycemic | 4839 (36.6) | 2723 (33.1) | 996 (46.8) | 598 (47.1) |
| Diabetic | 2413 (15.0) | 1098 (10.1) | 582 (24.4) | 456 (34.6) |
| **DM status at visit 2 N (%)** | | | | |
| Normoglycemic | 2462 (22.5) | 1854 (26.9) | 258 (13.6) | 98 (7.5) |
| Hyperglycemic | 3891 (28.6) | 2364 (27.4) | 757 (34.8) | 392 (31.5) |
| Diabetic | 2438 (14.7) | 1160 (10.5) | 584 (24.0) | 445 (31.0) |
| Missing DM status/ did not participate in visit 2 | 3551 (34.2) | 2185 (35.2) | 523 (27.6) | 335 (30.0) |
| **Center N(%)** | | | | |
| Bronx | 3221 (28.7) | 1965 (29.0) | 530 (27.1) | 292 (25.2) |
| Chicago | 2954 (15.1) | 1944 (16.6) | 508 (13.9) | 305 (13.8) |
| Miami | 3346 (31.8) | 1833 (27.9) | 552 (34.3) | 359 (36.3) |
| San Diego | 2821 (24.3) | 1821 (26.5) | 532 (24.7) | 314 (24.7) |
| **Alternative healthy eating index[a]** | | | | |
| Mean (SD) | 47.35 (7.30) | 46.96 (7.19) | 49.49 (7.41) | 49.58 (7.55) |
| **Shift work[b] N(%)** | | | | |
| No | 10,023 (79.4) | 6076 (78.2) | 1735 (80.4) | 1063 (83.2) |
| Yes | 2319 (20.6) | 1487 (21.8) | 387 (19.6) | 207 (16.8) |
| **Total physical activity[c]** | | | | |
| Mean (SD) | 712.47 (1079.68) | 747.51 (1077.77) | 644.78 (1002.69) | 605.72 (1106.28) |
| **Vigorous physical activity[d]** | | | | |
| Mean (SD) | 42.01 (95.75) | 43.54 (94.89) | 38.40 (92.85) | 35.26 (92.49) |
| **Smoking (%)** | | | | |
| Never | 7363 (60.6) | 4801 (64.9) | 1175 (53.5) | 648 (52.9) |
| Former | 2468 (17.4) | 1284 (14.4) | 548 (25.1) | 395 (28.7) |
| Current | 2498 (22.0) | 1471 (20.7) | 398 (21.4) | 226 (18.4) |
| **Alcohol use** | | | | |
| Never | 2403 (18.1) | 1444 (17.7) | 389 (16.4) | 244 (18.9) |
| Former | 4006 (29.7) | 2450 (29.3) | 711 (30.0) | 428 (33.3) |
| Current | 5929 (52.2) | 3666 (53.0) | 1022 (53.6) | 597 (47.8) |

Bold font represents variable names and column names.

*OSA* obstructive sleep apnea, categorized according to the respiratory event index (REI). No OSA: REI < 5. Mild OSA 5≤REI≤15, moderate-to-severe OSA: REI ≥ 15. *BMI* body mass index, *DM* diabetes mellitus.

[a]A measure of diet quality based on foods and nutrients predictive of chronic disease risk.

[b]Shift work was considered either afternoon, night, split, irregular, or rotating shift. Unemployed or dayshift is not shift work.

[c]The total amount of time spent doing some form of physical activity in a week (MET minutes/day) based on self-report.

[d]The average amount of time spent per day doing vigorous physical activity (MET minutes/day) based on self-report.

from the logistic regression based on MGB Biobank dataset (Supplementary Table 4 provides characteristics of the MGB dataset). Figure 1a shows the proportions of individuals by DM status and the change in their DM status over time, between visit 1 and 2 (6.03 years on average), stratified by mgbPRSsum T2D-PRS quartiles. This figure demonstrates that individuals with higher values of the PRS tend to have worse DM profiles (e.g., DM already at V1). Supplementary Fig. 1 shows distribution of the three new T2D-PRSs by DM category among visits 1 and 2 participants, demonstrating that, for the three PRSs, PRS distributions are shifted across individuals grouped by DM profiles (no/normal-glycemic, pre-DM/hypergelycemic, and DM).

Association analysis of T2D-PRSs with DM and incident DM in the overall dataset showed associations of all newly developed T2D-PRSs. Results across the three new T2D-DM PRSs were roughly similar. At baseline (Fig. 1b), per 1 standard deviation (SD) increase of the PRS, PRSsum was associated with increased visit 1 prevalence of DM with odds ratio (OR) = 3.13, 95% confidence interval (CI) [2.75; 3.56], gapPRSsum had OR = 2.77, 95% CI [2.44; 3.13], and mgbPRSsum had OR = 2.67, 95% CI [2.4; 2.97] for mgbPRSsum. Figure 1c shows corresponding results for incident DM ($N = 803$) among individuals with normal glycemia and hyperglycema. Incident rate ratios (IRRs) ranged from 2.02 (PRSsum) to 2.15 (gapPRSsum) per 1 SD increase of the PRS, and all associations were highly statistically significant. In contrast, while associations with PGS002308_PRS and PGS003867_PRS (see Table 1 for description of these PRSs) were also statistically significant, effect sizes were substantially lower, with ORs of 1.55 and 1.93 for baseline DM, and IRRs of 1.42 and 1.48 for incident DM. Supplementary Fig. 2 provides results from association analysis of the three new T2D-PRSs with DM and incident DM stratified by self-reported Hispanic/Latino background (characteristics of the HCHS/SOL target population stratified by background are provided in Supplementary Table 5). Here too, for each background group, the three PRSs had similar effect estimates, and all associations were statistically significant. Across groups, effect estimates from association analysis with baseline DM were highest for the South American group, with ORs of 4.44–5.63 (though wide confidence intervals given the sample size), and for incident DM the effect size estimates were highest for the Dominican group, with IRRs of 3.48–3.88.

Association analysis between the three T2D-PRSs with prevalent DM and incident DM stratified by OSA severity categories showed stronger associations in individuals with no or mild OSA compared to those with moderate-to-severe OSA, for both analyses (Fig. 1d; providing results for mgbPRSsum, Supplementary Fig. 3 provides results for the three T2D-PRSs). For example, in individuals with no OSA, mgbPRSsum had OR = 2.81, 95% CI [2.81; 3.30] for visit 1 prevalent DM and IRR = 2.2, 95% CI [1.77; 2.73] for incident DM, while in individuals with moderate-to-severe OSA it had OR = 2.16, 95% CI [1.68; 2.78] and IRR = 1.44 95% CI [1.11; 1.87] for visit 1 and incident DM, respectively. However, in interaction analysis models that included all individuals and interactions terms for T2D-PRSs (each in a separate model) with mild and with moderate-to-severe OSA, the interaction terms were not statistically significant, $0.21 < p < 0.88$ (Supplementary Fig. 4). We also report AUCs assessing the predictive performance of mgbPRSsum by comparing models with and without the PRS. While 95% ICs of the AUCs overlapped between models with and without the T2D-PRS, AUC values increased across all OSA severity strata and in the model using the complete population (Supplementary Fig. 5). For example, the AUC in models predicting incident DM (regardless of sleep phenotype) increased from 0.75 in the model without the T2D-PRS to 0.80 in the model with the PRS.

## Evidence that OSA mediates the effect of genetic determinants of T2D on DM

The T2D-PRSs were associated with increased risk of OSA. Specifically, per 1 SD increase of mgbPRSsum, the OR for mild-to-severe OSA (versus no

**Fig. 1 | T2D-PRSs associations with DM.**
**a** Proportion of individuals by DM status over time (between clinic's visit 1 and 2) stratified by T2D-PRS quartiles. The number of individuals represented by each tile is written on the tile. Note: "Persistent" refers to having the same DM status in both visit 1 and visit 2; "Worsen" refers to change in DM status from normoglycemic at visit 1 to hyperglycemic at visit 2 or from hyperglycemic in visit 1 to diabetic at visit 2; "Improve" refers to change in DM status from hyperglycemic at visit 1 to normoglycemic at visit 2 or from diabetic in visit 1 to hyperglycemic or normoglycemic at visit 2. **b** Estimated OR per 1 SD increase in T2D-PRSs in association with DM at baseline in HCHS/SOL individuals. **c** Estimated IRR per 1 SD increase in T2D-PRSs in association with incident DM in individuals free of DM at baseline. In **b** and **c** colors correspond to different PRSs, as denoted in the legend. **d** Association of 1 SD increased in mgbPRSsum with DM and incident DM in HCHS/SOL individuals stratified by OSA severity levels and overall dataset. Colors correspond to OSA strata, as denoted in the legend. All *p*-values were obtained from the 1 degree-of-freedom Wald test. OSA severity levels were defined based on the respiratory even index (REI): mild OSA was defined as 15 ≥ REI ≥ 5, moderate-to-severe OSA was defined as REI ≥ 15, and REI < 5 was considered no OSA. Throughout, error bars represent 95% confidence intervals. All models were adjusted for age, sex, BMI, study center and 5 genetic PCs. OR odds ratios, IRR incidence rate ratios, AUC Area Under the ROC (receiver operating characteristic) Curve, T2D type 2 diabetes, PRSs polygenic risk scores, DM diabetes mellitus, HCHS/SOL Hispanic Community Health Study/Study of Latinos, EDS excessive daytime sleepiness, OSA obstructive sleep apnea, SD standard deviation.

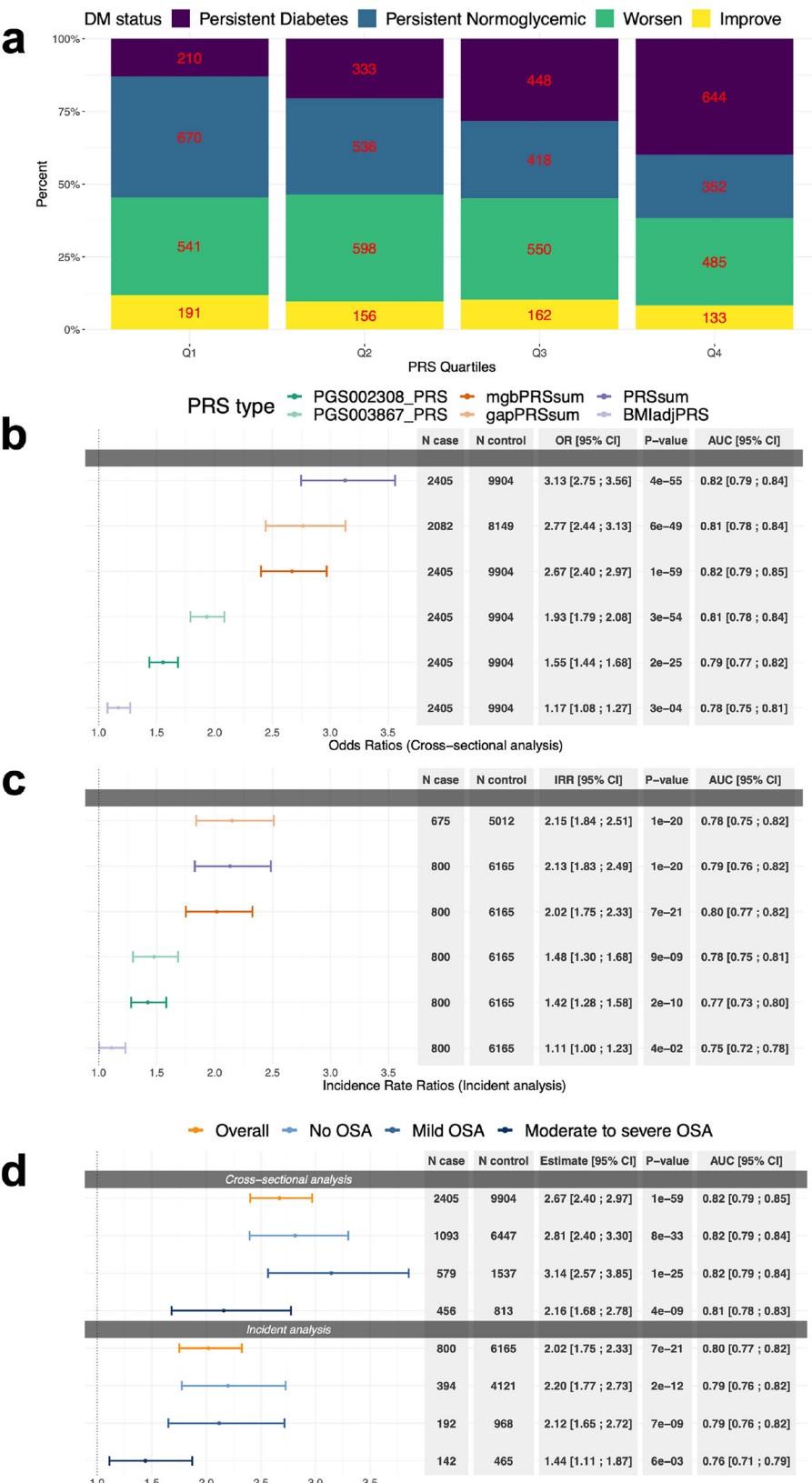

OSA) was 1.15, 95% CI [1.06; 1.26]. Associations with other sleep phenotypes were not statistically significant (Fig. 2a). Supplementary Fig. 6 shows results for the estimated association between all considered T2D-PRS (including the one constructed based on BMI-adjusted GWAS in individuals of European ancestry) and other sleep phenotypes. Results were consistent with those in Fig. 2a, with the exception that the BMIadjT2D-PRS did not appear to be associated with mild-to-severe OSA (OR = 1.04, *p* = 0.3). Adjusting for WHR instead of BMI resulted in a statistically significant association between all T2D-PRSs and mild-to-severe OSA (ORs = 1.1–1.17). Complete results are provided in the Supplementary

**Fig. 2 | Associations of T2D-PRSs with poor sleep phenotypes, estimated mediation effect by OSA, causal effects of T2D on OSA and OSA on T2D. a** Estimated OR per 1 SD increase of mgbPRSsum in association with poor sleep health at baseline in HCHS/SOL individuals. The comparison categories are individuals without the stated poor sleep phenotype (e.g., short sleep versus individuals who do not have short sleep, etc.). **b** Distribution of mgbPRSsum computed over all individuals with genetic data and DM in visit 2 ($N = 2483$), horizontal dashed lines denote quantiles of the PRS values $Q_0$–$Q_4$. **c** Estimated percents of risk mediation by mild-to-severe OSA in the association between T2D-PRS and incident DM in individuals who participated at the second visit to a clinic ($N = 6291$). Darker shades correspond to higher estimates. Estimates are provided for set values of the T2D-PRS, selected according to the distribution quantiles. Significance codes: 0 >= '***' <0.001 >= '**' <0.01 >= '*' <0.05 ' ' <0.1. **d** Estimated causal effect of T2D on OSA based on SNPs selected using $p$-value threshold $<5 \times 10^{-8}$ in BMI-adjusted and BMI-unadjusted T2D GWASs. **e** Estimated causal effect of OSA on T2D based on SNPs selected using $p$-value threshold $<5 \times 10^{-8}$ in BMI-adjusted and BMI-unadjusted OSA GWASs. Estimates are provided from a few MR methods (primary method: IVW, secondary: MR-PRESSO, and MR-RAPS), denoted by different colors. Results are not presented for MR-PRESSO in panel (**d**) for estimated effect of OSA on T2D in BMI adjusted analysis because they were the same as the results for the IVW method. Throughout, error bars represent 95% confidence intervals. All models were adjusted for age, sex, BMI, study center and 5 genetic PCs. T2D type 2 diabetes, OSA obstructive sleep apnea, IVW inverse variance weighted, BMI body mass index, AUC Area Under the ROC (receiver operating characteristic) Curve, SD standard deviation, SNPs Single-nucleotide polymorphism, GWAS genome wide association study

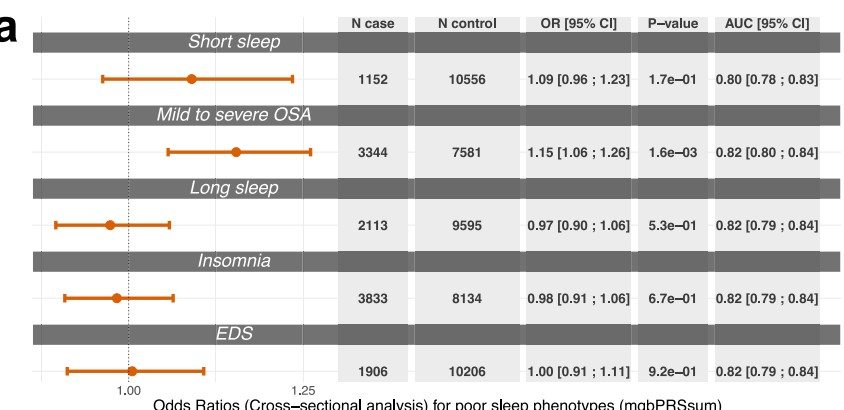

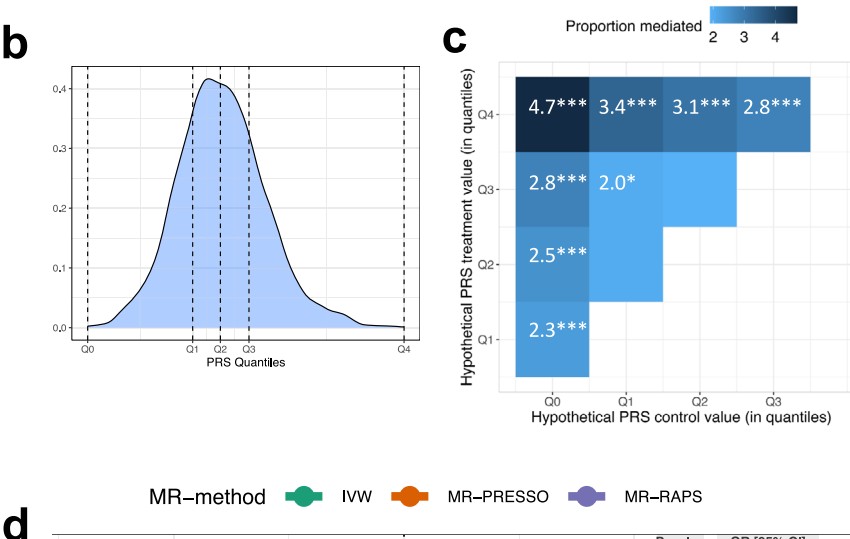

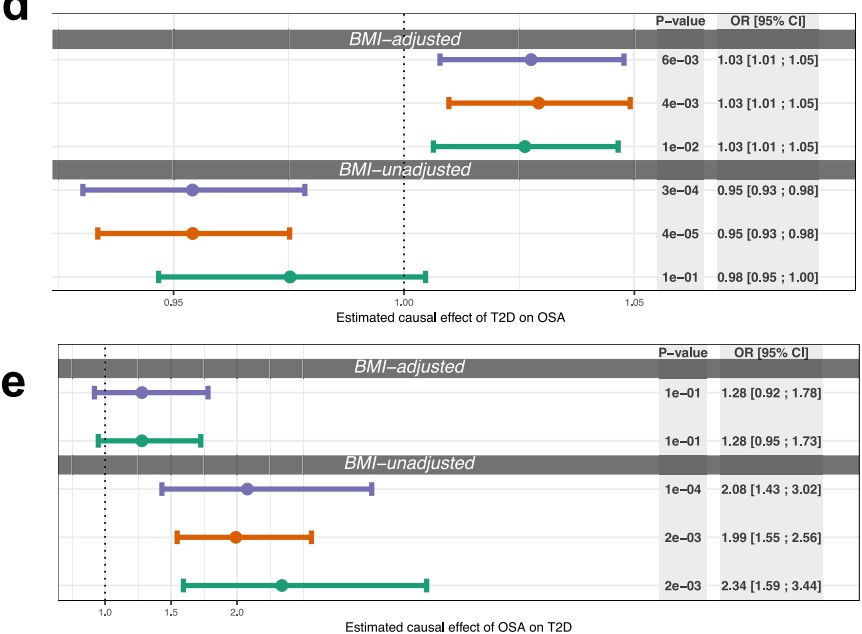

Table 6. Comparison of the results between BMI and WHR are visualized in the Supplementary Fig. 7. In another analysis, when adjusting for additional potential confounders (medication, socioeconomic status, physical activity and self-reported Hispanic/Latino background) the association between all T2D-PRSs and both mild-to-severe OSA and moderate-to-severe OSA remained about the same across analyses (Supplementary Tables 7 and 8).

In mediation analysis, OSA phenotypes mediated some of the T2D-PRS effect on DM development. Figure 2c (right panel) demonstrate the

estimated percent mediated effect when using mgbPRSsum as the exposure and mild-to-severe OSA as the mediator of incident DM (outcome). In the most extreme setting, the proportion of mediation by mild-to-severe OSA versus no OSA was 4.7% when assuming that in the population the mgbPRSsum value increases from the lowest (denoted as Q0 in the figure) to the highest (denoted as Q4) observed value in the sample (see Fig. 2b for a visual of the Q0-Q4 PRS values cut-off points). Supplementary Fig. 8 provides results from mediation analyses using mgbPRSsum as exposure and REI and moderate-to-severe OSA as mediators, showing consistent results with the primary analysis, though with lower estimated percentages of mediation. Supplementary Fig. 9 provides results from mediation analyses using BMIadjT2D-PRS as exposure and the three OSA phenotypes as mediators. When using REI and mild-to-severe OSA as mediators there were statistically significant estimates of percentages of mediated risk, but not when using moderate-to-severe OSA, potentially due to lower power given smaller number of individuals with moderate-to-severe OSA.

### Genetic analysis suggests causal effect of OSA on T2D

Given the mediation results, we estimated the causal effect of T2D on OSA, and of OSA on T2D, using two-sample MR based on summary statistics from published GWAS of OSA and of T2D (Table 2). We identified 142 and 182 SNPs that passed the filtering steps to serve as instruments for MR of T2D on OSA based on BMI-adjusted and BMI-unadjusted T2D GWASs. Figure 2d provides results from MR of T2D on OSA, showing evidence for a weak causal association of T2D on OSA when accounting for BMI via BMI-adjusted analysis: IVW OR = 1.03, 95% CI [1.01; 1.05]; the same effect sizes were observed in sensitivity analyses using other MR methods. BMI-unadjusted anlaysis suggested protective or no causal effect: IVW OR = 0.98, 95% CI [0.95; 1].

In MR analysis of OSA on T2D, we had 2 and 10 SNPs serving as instruments for MR based on BMI-adjusted and BMI-unadjusted primary analyses. Results are visualized in Fig. 2e. BMI-unadjusted MR suggested causal effect of OSA on T2D (IVW OR = 2.34, 95% CI [1.59; 3.44]) but in BMI-adjusted analysis the association weakened and was no longer statistically significant. Supplementary Fig. 10 demonstrates similar results based on an analysis that utilized weak instruments ($p < 10^{-7}$ in the OSA GWAS). Supplementary Data 1 provide complete lists of SNPs used in estimation of causal effect of T2D on OSA and of OSA on T2D.

Because MR analysis was limited by the availability of GWAS of only European ancestry populations, and only a few strong instruments for OSA (reducing power), we also constructed OSA-PRSs in HCHS/SOL and (a) validated its association with OSA, and then (b) estimated its association with DM (noting that the interpretation of such an association is limited to prediction rather than to causality). Supplementary Fig. 11 demonstrates that the two OSA-PRSs (BMIadjOSA-PRS and BMIunadjOSA-PRS, based on BMI-adjusted and -unadjusted GWAS, respectively) were associated with moderate-to-severe OSA versus no-and-mild OSA, as well as with mild-to-severe OSA versus no OSA. The estimated associations of two OSA-PRSs with baseline DM were similar and close to null for both BMIadjOSA and BMIunadjOSA PRSs (OR = 1.03, 1.02) and statistically insignificant ($p > 0.7$). These results are provided in Supplementary Fig. 12. Note that all association analyses were BMI-adjusted.

We also used MVMR in an attempt to estimate the causal effects of OSA and of T2D while adjusting for BMI as a second exposure. Thus, MVMR used a BMI-unadjusted GWASs. The analysis suggests a modest protective causal effect of T2D on OSA (OR = 0.97, 95% CI [0.95; 0.98]), similar to the BMI-unadjusted MR analysis of T2D causal effect on OSA. In the reverse direction, OSA showed a strong causal effect on T2D (OR = 2.01, 95% CI [1.85; 2.18]). The results are provided in Supplementary Table 9.

### Discussion

As a summary of the key clinical insights of this work: In a large Hispanic/Latino cohort using multi-ancestry T2D-PRS designed to be powerful for individuals from admixed ancestral backgrounds, the data revealed that OSA is both a causal risk factor for DM and also mediates some of the

genetic risk for developing DM. This suggests that OSA is a modifiable risk factor for both existing and incident cases of DM. More specifically, we developed multi-ancestry T2D-PRSs and used them to study the relationship between OSA and DM in the HCHS/SOL. T2D-PRSs were highly associated with DM, and, contrary to our expectation, their association with DM appeared weaker in individuals with moderate-to-severe OSA (estimated OR = 2.16) compared to individuals with no (OR = 2.81) or mild (OR = 3.14) OSA. However, the interaction test was not statistically significant. Subsequent analysis suggested that a small proportion of the T2D-PRS effect on DM is mediated via an increase in OSA severity. Causal association analysis using two-sample MR and MVMR suggested that OSA causally increases risk of T2D, while MR analysis with T2D as the exposure had results that are difficult to interpret.

We used the PRS-CSx package to develop ancestry-specific PRSs, which we then combined as sums: PRSsum, gapPRSsum, and mgbPRSsum. The three PRSs had similar performance overall (with small variations across various analyses and stratifications) and additional datasets are needed to potentially identify whether one approach is better than other approaches. Critically, gapPRSsum used individual-specific weights, depending on the estimated global ancestry proportions of an individual to sum the PRSs. The number of individuals for whom gapPRSsum ($N = 10,258$) was computed is lower than the sample sizes used in analysis with the other PRSs ($N = 12,342$), due to missing availability of estimated genetic ancestry proportions. To generalize the gapPRSsum to other populations, one needs to have both ancestry-specific PRSs and the corresponding ancestry proportion estimates for each individual in the new population. This is a limitation because these may not be available, or the specific ancestry selected (or most appropriate) for inference for a given population may not correspond to the ancestry-specific PRSs. Further, the same strategy may not be applicable for different ancestry specifications due to lack of data availability. Other PRS combination strategies may also be limited: mgbPRSsum is limited due to the use of the specific MGB population, which has different ancestry composition (and other demographic characteristics, that may impact PRS weights estimation) than other target populations. PRSsum is limited in that it places the same weight on all PRSs, while one or some may be substantially stronger than others due to the sample size in the source GWAS population. More generally, it is plausible that different PRS combination weights are suitable for different populations, just like different weights may be suitable for different individuals (as in gapPRSsum). The sample size in our study likely was too small to allow for reaching a conclusion either way, and ideally, several studies of well-defined genetic ancestry makeup would be used to assess ideal weighting strategies. In summary, the three PRSs had good performance and improved over existing PRSs, and either one can be used for future research. Finally, another limitation of gapPRSsum is that 1000 Genome AMR reference panel does not perfectly matches the Amerindian ancestry proportions estimated in HCHS/SOL, as the latter correspond to homogeneous Amerindian (Native American ancestry), while the 1000 Genome AMR reference population includes admixed American individuals.

Several earlier studies, including those conducted in Hispanic/Latino individuals, demonstrated evidence of the association between OSA and other measures of poor sleep health with DM[11,12,15,56–59]. Here we observe a strong association between T2D-PRSs and OSA. This is consistent with OSA being a risk factor for T2D: genetic variants underlying T2D risk factors should be associated with T2D. Thus, we do expect that genetic risk of T2D would be associated with OSA as it is a risk factor of T2D. In this case, a mediation effect of T2D PRS on T2D mediated via OSA also makes sense. Indeed, we estimated a statistically significant mediation effect by OSA on the association between T2D-PRS and incident DM, albeit modest. This mediation effect further supports the weaker T2D-PRS and DM association in the more severe OSA category. This is because an analysis restricted to individuals with mild-to-severe OSA only estimates the portion of the T2D-PRS on DM that is not mediated via mild-to-severe OSA, i.e., only the direct effect and not the total effect. However, there was no statistically significant evidence of the association of T2D-PRS with insomnia, short sleep, long

sleep, or EDS. It is possible that these sleep measures are not strong risk factors of T2D, but rather have shared common causes with T2D, for example. A meta-analysis of published studies examined the associations between OSA with pre-DM and DM, including the impact of the severity of OSA on DM. The results showed that OSA is associated with a higher risk of DM, and DM-related glycemic traits, both longitudinally[60–62] and cross-sectionally. Furthermore, results from previous studies have shown a bidirectional association between T2D and OSA[8,16,63,64]. Here, we, too, observed a likely causal association of OSA on T2D in MR and MVMR analysis. The results are more complicated for the reverse causal association: T2D had an estimated protective on OSA in MR analysis that was not adjusted for BMI, and also a statistically significant protective association in MVMR with BMI as an adjusting covariate (modeled as a second exposure in the MR). A protective association of T2D on OSA is not supported by the literature. In MR analysis based on GWAS summary statistics that were BMI-adjusted, T2D had a risk increasing association with OSA. To assess these results, we consider the MR literature. Specifically, a study by Hartwig et al.[65] reported a comprehensive simulation study investigating the estimated causal effects from two-sample MR when an exposure and outcome association may have various types of confounders, and when using GWAS summary statistics that were and were not adjusted for an important confounder[65], such as BMI. This manuscript indeed had settings that resulted in patterns that we see in the T2D-OSA univariate MR analysis, i.e., where T2D has a protective estimated effect on OSA in BMI-unadjusted analysis, and risk increasing in BMI-adjusted analysis. Based on the simulations in Hartwig et al., we think that a causal structure that may results in such change in direction of the estimated causal effect in univariate MR analysis, is having genetic variants that are common causes of both T2D and BMI (independently, not with BMI mediating their effect on T2D), and in addition having an existing unmeasured common cause of BMI and OSA (see Figure 4 in Hartwig et al.). This is a complicated causal structure, and it is difficult to more specifically hypothesize what the unmeasured confounder that is a common cause of both BMI and OSA. A reasonable hypothesis may be that variants that influence specific fat depots, e.g., visceral or tongue adipose tissue, could be common causes of both traits given recent results showing associations of OSA PRS being associated with adipose tissue distribution measures[66]. Future work will need to study MVMR estimation performance in this causal structure (a recent paper applying MVMR to resolve biases caused by covariate adjustment in GWAS did not consider this causal structure[67]), and perhaps use causal structural equation models to (genomic SEMs) to further study the causal structure behind these phenotypes[68].

While some earlier MR-based investigations of the causal effect of OSA on T2D reported no direct causal effect by OSA[69,70], others were able to establish a likely association[71] suggesting the potential indirect impact of OSA on DM via BMI. These results support our findings in two-sample MR, which used published summary statistics from GWAS in individuals of European genetic ancestries, rather than the HCHS/SOL dataset. In the MR analysis potential causal effect of OSA on T2D was found in univariate MR BMI-unadjusted analysis and in MVMR adjusted for BMI.

Previous publications studied the associations between sleep-related traits (including sleep duration and quality, insomnia, short and long sleep, morningness-eveningness chronotype, and others) and risk of DM. For example, others[72–74] used MR analysis and established causal association between insomnia, chronotype-related gene expression, and DM, but found no evidence of a causal association between sleep duration and the risk of DM. Also, published analyses reported statistically significant genetic correlations of T2D with insomnia, and short and long sleep[75]. In contrast, our results, using a different analytic approach, showed weak associations between T2D-PRS and insomnia, and short and long sleep. However, only modest sample sizes and potential confounding with other health measures and health behaviors may have limited our power. For example, individuals with healthy sleep are younger individuals than others, and this may substantially affect the association results. Differences in questionnaires and measurement methods used to assess sleep phenotypes may also contribute to differences in findings between studies.

A strength of this work is that we constructed multi-ancestry T2D-PRS that factor information on an individual's admixed ancestry and may be more suitable in analyses that are applied to samples of Hispanic/Latino individuals. Previously, it was demonstrated that PRSs are less effective in predicting outcomes for individuals whose genetic background substantially deviates from that of the participants in the original GWAS from which the scores were calculated[76,77]. Thus, accounting for multiple genetic ancestries in the construction of multi-ancestry T2D-PRSs potentially enhances predictive power in the analysis performed on the data of Hispanic/Latino adults. Further, our study is based on the population of diverse Hispanic/Latino adults in the U.S., which is unrepresented in research. OSA was assessed via an objective, overnight sleep study, and we used a number of analytic strategies, leveraging both individual-level data (PRS, mediation analyses) and summary statistics (MR) to study how DM and OSA related to each other (and other sleep phenotypes in secondary analyses). A limitation in this study is that we treated type 1 and type 2 DMs combined as DM. We were unable to distinguish between the two types of DM because the HCHS/SOL dataset only contains information on a DM status without specifying DM type. Nevertheless, given the low prevalence of T1D and the strong associations of the T2D-PRSs with both baseline DM and incident DM, which is likely mostly T2D given that our study population was, for the most part, older than age 18 years at baseline, we believe that the results are robust, in that T2D-PRSs are predictive and the relationship with OSA is well characterized.

In summary, we developed PRSs for T2D utilizing information from multiple genetic ancestries and specifically focusing on the ancestral populations of admixed Hispanic/Latino individuals in the U.S. The multi-ancestry T2D-PRSs had a strong association with baseline DM and incident DM. They were also associated with OSA, providing evidence that OSA mediates some of the T2D risk conferred by underlying genetic factors, presumably those underlying OSA. This work extends our current knowledge of understanding the effect of OSA on the risk of developing DM later in life, given an individual's genetic predisposition to T2D.

## Data availability
Ancestry-specific summary statistics from GWAS of DM published in 2022 (Mahajan et al.) in the DIAGRAM consortium were downloaded from https://diagram-consortium.org/downloads.html. HARE-group specific summary statistics from GWAS of DM in MVP published in 2020 (Vujkovic et al.[33]) were downloaded by dbGaP application to study accession phs001672. SNPs and weights for ancestry-specific T2D-PRSs are provided in the GitHub repository https://github.com/YanaHrytsenko/DM_PRS_OSA_mediation[55]. The source data for Figs. 1 and 2 is in the Supplementary Data 3. HCHS/SOL data are available through application to the data base of genotypes and phenotypes (dbGaP) accession phs000810, or via a data use agreement with the HCHS/SOL Data Coordinating Center (DCC) at the University of North Carolina at Chapel Hill, see collaborators website: https://sites.cscc.unc.edu/hchs/. MGB Biobank data are available to MGB investigators via a web-based portal.

## Code availability
Code used for analysis in this paper is publicly available on the GitHub repository https://github.com/YanaHrytsenko/DM_PRS_OSA_mediation[55].

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

## Acknowledgements

The authors thank the staff and participants of HCHS/SOL for their important contributions. We thank Mass General Brigham Biobank for providing samples, genomic data, and health information data. The work was supported by National Heart Lung and Blood Institute (NHLBI) grants R01HL161012 to T.S. and R35HL135818 to S.R., and National Institute on Aging grant R01AG080598 to T.S. The Hispanic Community Health Study/Study of Latinos is a collaborative study supported by contracts from the National Heart, Lung, and Blood Institute (NHLBI) to the University of North Carolina (HHSN268201300001I/N01-HC-65233), University of Miami (HHSN268201300004I/N01-HC-65234), Albert Einstein College of Medicine (HHSN268201300002I/N01-HC-65235), University of Illinois at Chicago (HHSN268201300003I/N01- HC-65236 Northwestern Univ), and San Diego State University (HHSN268201300005I/N01-HC-65237). The following Institutes/Centers/Offices have contributed to the HCHS/SOL through a transfer of funds to the NHLBI: National Institute on Minority Health and Health Disparities, National Institute on Deafness and Other Communication Disorders, National Institute of Dental and Craniofacial Research, National Institute of Diabetes and Digestive and Kidney Diseases, National Institute of Neurological Disorders and Stroke, NIH Institution-Office of Dietary Supplements.

## Author contributions

Y.H. and T.S. drafted the manuscript. Y.H., B.W.S. and T.S. performed data analysis. M.L.D., L.C.G., C.R.I., J.C., and S.R. contributed to study design and data curation in HCHS/SOL. H.W., S.B., K.D.T., O.G., A.R., M.L.D., L.C.G., C.R.I., J.C., Q.Q., C.A., and S.R. critically reviewed the manuscript.

## Competing interests

Dr. Redline discloses consulting relationships with Eli Lilly Inc. Additionally, Dr. Redline serves as an unpaid member of the Apnimed Scientific Advisory Board, as an unpaid board member for the Alliance for Sleep Apnea Partners and for the National Sleep Foundation. All other authors declare no competing interests.
