## [Transparent Peer Review file · Communications Medicine]

Obstructive sleep apnea mediates genetic risk of Diabetes Mellitus in the Hispanic Community

Corresponding Author: Dr Tamar Sofer

Version 0:

Reviewer comments:

Reviewer #1

(Remarks to the Author)

Hrytsenko and colleagues report a study where they constructed a multi-ancestry PRS for T2DM using data from the DIAGRAM consortium and MVP. They used this PRS to study the association of a multi-ancestry PRS for T2DM among Hispanic/Latino individuals (as part of HCHS/SOL cohort), the association of this PRS with OSA, and how much of the polygenic susceptibility for T2DM is mediated by OSA. They additionally performed a bi-directional 2-sample MR study to study the potentially causal link between T2DM and OSA. Authors report that their new multi-ancestry PRS for T2DM is associated with both prevalent and incident DM, OSA, and a small portion of this genetic susceptibility (at highest 4.7%) is seemingly mediated by baseline OSA. Authors also show evidence towards a bi-directional causal link between T2DM and OSA (much larger in effect in the direction of OSA -> T2DM). In general, BMI-adjusted analyses revealed decreases in the observed association metrics and loss of statistical significance.

In general, these results raise the hypothesis that BMI (which is significantly associated with both T2DM and OSA) might be the primary factor linking these two diseases, despite the brief discussion of all possible mechanisms/pathways mentioned in the manuscript connecting T2DM and OSA. I build this impression on the fact that BMI-adjusted analyses decrease the strength of association/make them non-significant, differences in the effect size in bi-directional MR analyses (OSA -> T2DM much larger), and our understanding of T2DM and OSA pathogenesis from prior literature. It would be helpful to repeat PRS association analyses (PRS constructed without BMI adjustment in GWAS summary statistics for reasons that will be discussed below) with OSA, adjusted for baseline BMI.

Analyses that use BMI-adjusted GWAS summary statistics, especially in MR analyses, bring on added complexity due to possible collider bias in these SNP-level associations (e.g., a SNP that is associated with BMI but not with T2DM can appear to be significantly associated with T2DM when adjusted for BMI in the analyses, i.e., collider bias). To avoid this, authors should consider utilizing multivariable MR (in bi-directional analyses) for the association of T2DM and OSA, including BMI.

With respect to the mediation effect, this is quite small, even in the most extreme case (going from Q0 to Q4, as depicted in the results/figures). Quite honestly, I am not sure how to interpret this mediation; a PRS combines effects from a large number of variants, potentially of pleiotropic effects (especially in the context of T2DM; including the ones involved in energy metabolism, adiposity traits, possibly causing OSA through different pathways). Meaning, genetic risk, as quantified by a PRS for T2DM, will never be not associated with related risk factors. I am not sure what this adds mechanistically or clinically to our understanding of the development of T2DM. I would value MR analyses much more highly than these PRS analyses. Authors should justify the reasoning and implications of PRS analyses performed in this study.

Additional comments:

- Please double check the affiliations for any typos (e.g. Line 19)
- Line 56: Do authors mean T2D, instead of T2D-PRS, given these refer to the MR analysis results?
- Line 59: Conclusion on abstract, in its present form, does not capture the results effectively I think. Within the limitation of character constraints, should include that fact that there is a most mediation of polygenic risk and evidence of a potentially bi-directionally causal link between T2DM and OSA.
- Lines 80-82: It would be helpful to briefly expand on these potential pathways given it is not readily apparent to the reader
- Lines 95-97: Authors indicate that a prior analysis in HCHS/SOL using 2-sample MR showed DM and glycemc traits

exhibit a causal association with OSA but not the other way around. Then authors, in the following sentence, go on to say that these results indicate the existence of a potential bidirectional association between OSA and diabetes. This is seemingly clarified between the lines 99-104, however rewording can avoid the confusion further.

- Lines 221-224: What reference panel was used to infer these four ancestral populations? If it is 1000 genomes, AMR would stand for 'admixed american' and these include Mexicans and Puerto Ricans from the US, Colombians, and Peruvians (thus, although including an admixed Hispanic/Latino population, not really Native American [Amerindian] i.e., a homogenous continental ancestry group).

- Lines 227-230: Were strand ambiguous variants removed prior to PRS construction? Please clarify.

- Lines 232-233: Were PRS standardized to mean 0 and variance 1 prior to each analyses? If not, regression betas will not exactly correspond to increase in trait/odds per 1 SD increase in PRS. Please clarify and revise as needed.

- Lines 257-258: This is an unusual way of obtaining CIs for AUC. Why did authors not use bootstrapping as commonly done in the field? Please justify.

- Lines 310-312: This sentence is written a bit informally, especially the latter part, please consider revising.

- Line 461: This subsection title is a very bold claim. MR analyses, although robust to some of the challenges of observational studies, are not infallible given its strong assumptions (some of which are not verifiable despite all sensitivity analyses and other advances in methodology). Please temper the language to reflect that (such as '... evidence that T2D potentially has a causal association with OSA' or similar). I gave line 461 as an example, however, all MR analysis results discussed in the text should reflect this.

- Lines 501-504: Authors should also include here that the size of this effect, in addition to not being significant, was also quite modest, in order to avoid unintentionally misleading the readership.

- Lines 504-505: Again, quite modest percentage of mediation, please be clear to communicate this.

- Figure 1 color legends for panels b,c, and d are not clear. I am honestly not sure what I am looking at in terms of groups for ORs. Please revise the figure to make it very clear what OR corresponds to which analysis/group.

- Figure 2: Please increase the point size for the dots denoting the OR value. The way it is right now, it almost looks like just a horizontal line depicting the CIs.

Reviewer #2

(Remarks to the Author)

This study systematically analyzes how certain components of the polygenic risk score (PRS) for type 2 diabetes mellitus (T2DM) may mediate the relationship between obstructive sleep apnea (OSA) and the incidence of T2DM. The background, methodology, and results are well-explained and clearly articulated throughout the manuscript. Additionally, the authors' approach of addressing the challenge of a mixed population by comparing and applying various PRS methodologies is commendable and demonstrates a thoughtful strategy. Below, I have provided comments on areas that could benefit from further refinement:

Comments: : Figures 1 and 2 - Bar Chart Formatting: The bar lines in Figures 1 and 2 could be slightly thicker to enhance clarity and visual distinction between data points. Additionally, the legends appear too small, making them difficult to read. Adjusting the font size of the legends to improve readability would greatly aid in understanding the charts.

Reviewer #3

(Remarks to the Author)

This manuscript investigates the association between diabetes and obstructive sleep apnea in the Hispanic/Latino population, utilizing multi-ancestry type 2 diabetes polygenic risk scores (T2D-PRSs). The findings are of interest and the study addresses an important question; however, there is room for improvement in the clarity and presentation of the results, particularly regarding the figures and associated descriptions.

Comments:

1. In Introduction section, the sentence "it is possible that there is a more complex association between OSA and risk of diabetes" is somewhat ambiguous, as it is not clear what the comparison is being made against. Clarifying what is meant by "more complex" could improve the reader's understanding.

2. The final sentence of the second paragraph in Introduction, ("OSA may modify genetic risk of diabetes") feels somewhat abrupt and underdeveloped. Since the following paragraph discusses genetic risk in more detail, this sentence may be unnecessary or better integrated with the next section to maintain a more coherent flow.

3. In the second sentence in the third paragraph of the Introduction, it may be helpful to clarify that intermittent hypoxia and fragmented sleep are characteristic consequences of OSA.

4. In the third paragraph of the Introduction, the authors refer to genetic studies showing interactions between sleep duration and genetic loci in relation to blood pressure and lipid levels, suggesting that similar mechanisms might underlie associations between OSA and diabetes. To strengthen the rationale, the authors might consider noting whether previous genetic studies have reported interactions between sleep-related traits and glycemic outcomes or diabetes risk. If such evidence exists, its inclusion would further support the study's premise.

5. In the Methods section, diabetes mellitus (DM) is defined according to the American Diabetes Association (ADA) criteria. In addition, DM status was also ascertained using scanned medication data at baseline and self-reported medication use at the second exam. It would be helpful to clarify what specific medications were considered indicative of DM in this context. Is there a possibility that some individuals who did not meet the ADA criteria may have been using diabetes medications for other purposes, such as early intervention, treatment of other conditions, or off-label use (e.g., for weight loss)? Furthermore, it would strengthen the rigor of the study to report how many participants were classified as having DM based on ADA criteria versus medication data. In addition, how many participants underwent blood tests, and what were the reasons for

missing data, if any, in participants who did not? Also, the sentence stating that "ADA criteria rely on laboratory tests, either [...], or scanned/transcribed anti-diabetic medication use" may be misleading. The use of anti-diabetic medications is not formally included in the ADA diagnostic criteria.

6. In the Results section, the authors refer to "hyperglycemic" participants; however, no definition of hyperglycemia is provided in the Methods section. To improve clarity and reproducibility, it is recommended that the criteria used to define hyperglycemia be explicitly stated in the Methods.

7. To enhance the interpretability and standalone value of the figures (including figures in supplementary material), it is recommended that the reference group or scale for both the ORs and IRRs be clearly specified in each figure and its corresponding caption. Currently, it is not immediately clear from the figures what these estimates reflect.

8. In Figure 1, panels b and c do not include legends identifying the plotted groups or lines, which makes interpretation difficult. In panel d, although a legend is provided, each group appears to have two separate plotted values. Based on the caption, it can be inferred that these represent DM and incident DM, but it is not clearly labeled which is which. To improve clarity and allow for independent interpretation of the figure, it is recommended that all panels include appropriate legends and that the meaning of the plotted values be clearly indicated, particularly in panel d.

9. The figure captions, use the phrasing "OR of the T2D-PRSs" and "IRR of the T2D-PRSs," which may be misleading, as they suggest the PRS is the outcome. To avoid confusion, it would be clearer to rephrase as "OR for DM" or "IRR for incident DM" to reflect that DM is the outcome and PRS is the predictor variable.

10. In Figure 2a, the addition of a label or legend would improve interpretability, as it is not immediately clear what the plotted data represent. Additionally, in panels c and d, although labels are present, each group appears to be plotted twice, and the distinction is not clearly labeled in the figure itself. To improve clarity, it is recommended that all plotted elements be explicitly identified, so that the figure can be understood independently of the main text.

Reviewer #4

(Remarks to the Author)

Dear Editor

I am grateful for the invitation and opportunity to review the manuscript "COMMSMED-24-1357-T Obstructive sleep apnea mediates genetic risk of Diabetes Mellitus: The Hispanic Community Health Study/Study of Latinos". The manuscript assessed the genetic risk between obstructive sleep apnea and diabetes.

In general, the manuscript presents consistent content and structure. Reading, vocabulary and interpretation are easy to understand. The methodology seems appropriate for the proposed objectives and the results are compatible with the proposed objectives. In addition, the discussion presents classic and current studies, thus generating new ideas for future studies.

I begin this review by contextualizing the role of variables related to lifestyle in the regression model to assess the strength of association. Dietary habits such as the absence of industrialized and ultra-processed foods, sleep patterns and the sleep-wake cycle no longer modulated by the "moon-sun" cycle but recently modulated by electrical energy, pedestrian commuting habits, among others, very common in the past, were very different from the current lifestyle. Thus, exploring the influence of these variables on the magnitude of the strength of association between OSA and diabetes may be timely and provide insights for understanding complex associations such as OSA and metabolic profile. Based on this comment, I suggest:

Consider other measures of obesity in the regression model

1.1 The authors used BMI as an obesity metric. Considering the role of visceral fat in mediating the results. Would replacing BMI with a measure of central obesity such as abdominal circumference or waist-to-hip ratio potentially provide more robust evidence on the strength of association in the model?

2 - Potentially confounding variables:

2.1- Medications: Did the participants regularly use medications known for their anti-inflammatory effects such as statins?

2.2- Socioeconomic status

2.3 - Physical activity: an important aspect to be included in the model, since from an evolutionary point of view, humans actively allocated their energy expenditure through walking, unlike the current sedentary behavior

3- Race - ancestry in the genetic model: Could one of the limitations of the model be the fact that Latinos also have indigenous ancestry?

STRENGTHS OF THE STUDY:

1 - the evaluation of the association between OSA and diabetes from the point of view of pleiotropy brings new information to the understanding of the complex metabolic imbalance of this mechanistic cascade of pathophysiological events. This approach is relatively recent and additive to information from cross-sectional studies.

2 – The epidemiological characteristic, the robustness of the association measures.

No further comments

Version 1:

Reviewer comments:

Reviewer #1

(Remarks to the Author)

Authors addressed all my comments. Thank you for their diligent work on the revision.

Reviewer #3

(Remarks to the Author)

The authors have appropriately addressed the issues raised in the previous review. No further revisions are necessary.

Reviewer #4

(Remarks to the Author)

Dear Authors

This manuscript comprehensively assessed the association between diabetes and obstructive sleep apnea. More than that, it deepened knowledge and provided new evidence. In an unprecedented way, it shows the cause-and-effect relationship between OSA and diabetes. From a research perspective, it sheds light on knowledge and fills an important gap in the technical-scientific literature, since to date the body of evidence is based largely on cross-sectional and longitudinal studies. From a clinical perspective, the findings of the present study also highlight the metabolic imbalances refractory to the presence of OSA. Together, these facts can potentially change decision-making for the treatment of OSA and associated conditions.

The authors' comments and justifications after the review further improved the initial manuscript significantly.

Response to review of “Obstructive sleep apnea mediates genetic risk of Diabetes Mellitus: The Hispanic Community Health Study/Study of Latinos”

Reviewer #1 (Remarks to the Author):

Hrytsenko and colleagues report a study where they constructed a multi-ancestry PRS for T2DM using data from the DIAGRAM consortium and MVP. They used this PRS to study the association of a multi-ancestry PRS for T2DM among Hispanic/Latino individuals (as part of HCHS/SOL cohort), the association of this PRS with OSA, and how much of the polygenic susceptibility for T2DM is mediated by OSA. They additionally performed a bi-directional 2-sample MR study to study the potentially causal link between T2DM and OSA. Authors report that their new multi-ancestry PRS for T2DM is associated with both prevalent and incident DM, OSA, and a small portion of this genetic susceptibility (at highest 4.7%) is seemingly mediated by baseline OSA. Authors also show evidence towards a bi-directional causal link between T2DM and OSA (much larger in effect in the direction of OSA -> T2DM). In general, BMI-adjusted analyses revealed decreases in the observed association metrics and loss of statistical significance.

Response: Thank you for your careful review!

In general, these results raise the hypothesis that BMI (which is significantly associated with both T2DM and OSA) might be the primary factor linking these two diseases, despite the brief discussion of all possible mechanisms/pathways mentioned in the manuscript connecting T2DM and OSA. I build this impression on the fact that BMI-adjusted analyses decrease the strength of association/make them non-significant, differences in the effect size in bi-directional MR analyses (OSA -> T2DM much larger), and our understanding of T2DM and OSA pathogenesis from prior literature. It would be helpful to repeat PRS association analyses (PRS constructed without BMI adjustment in GWAS summary statistics for reasons that will be discussed below) with OSA, adjusted for baseline BMI.

Response: We thank the reviewer for this helpful comment and would like to clarify that the proposed analysis was already performed. Specifically, in our primary analysis we developed three types of T2DM-PRSs (PRSsum, mgbPRSsum, gapPRSsum) using BMI-unadjusted multi-ancestry T2D GWASs, DIAGRAM and MVP, and performed the association analysis, adjusting for BMI as a covariate, between these T2DM-PRSs and several OSA severity categories, other sleep phenotypes categories. Additionally, we developed BMIadjPRS, based on the European BMI-adjusted T2DM GWAS, DIAGRAM, and performed similar association analysis as described

above. You can find more complete details in the “Table 1: Outline of the analyses performed in this study.” under the “PRS associations with poor sleep phenotypes at baseline.” section of the table.

Analyses that use BMI-adjusted GWAS summary statistics, especially in MR analyses, bring on added complexity due to possible collider bias in these SNP-level associations (e.g., a SNP that is associated with BMI but not with T2DM can appear to be significantly associated with T2DM when adjusted for BMI in the analyses, i.e., collider bias). To avoid this, authors should consider utilizing multivariable MR (in bi-directional analyses) for the association of T2DM and OSA, including BMI.

Response: Thank you for this suggestion. We address this comment by implementing the multivariable MR (MVMR) in bi-directional analyses for the association of T2DM and OSA including BMI as additional exposure. Specifically, we used the BMI-Unadjusted GWASs for T2DM, OSA and BMI and performed the MVMR analysis using TwoSampleMR R package. To compare results to the MR analysis using BMI-Adjusted GWASs data we performed the MVMR analysis using significant SNPs based on three different thresholds. The MVMR results suggest a protective causal effect of T2D on OSA in a BMI-adjusted analysis (OR=0.97), similar to the MR analysis based on BMI-unadjusted GWAS, and a strong risk-increasing causal effect of OSA on T2D (OR=2.01 when using genome-wide significant instruments). We updated the manuscript to report these results in secondary analysis. Importantly, your comment really helped us understand biases in MR due to adjustment better than before, and improve our interpretation of the MR results.

The results from MVMR investigating OSA causal effect on T2D adjusted to BMI were in our hypothesized direction. However, the results from MVMR investigating T2D causal effect on OSA adjusted to BMI, are unexpected (protective effect, as the MR results from BMI-adjusted GWAS). We reviewed related literature (MVMR, biases due to adjustment for a heritable covariate), and we still didn't find a satisfactory explanation for this. Focusing on the T2D -> OSA scenario, a very nice paper about MR when using adjusted GWAS covered practically all causal scenarios (Hartwig et al., “Bias in two-sample Mendelian randomization when using heritable covariable-adjusted summary associations”) while a recent paper about MVMR (Gilbody et al., “Multivariable MR Can Mitigate Bias in Two-Sample MR Using Covariable-Adjusted Summary Associations”) reviewed some, but not all scenarios of potential causal structures. Based on the simulations in Hartwig et al, we think that a causal structure that may result in such a bias in MR analysis is having genetic variants that are common causes of both T2D and BMI (independently, not with BMI mediating their effect on T2D), and in addition having an existing unmeasured

common cause of BMI and OSA. However, future work will need to study MVMR estimation performance in this causal structure.

We now write about this in the discussion (lines 615-641):

“Here, we, too, observed a likely causal association of OSA on T2D in MR and MVMR analysis. The results are more complicated for the reverse causal association: T2D had an estimated protective on OSA in MR analysis that was not adjusted for BMI, and also a statistically significant protective association in MVMR with BMI as an adjusting covariate (modeled as a second exposure in the MR). A protective association of T2D on OSA is not supported by the literature. In MR analysis based on GWAS summary statistics that were BMI-adjusted, T2D had a risk increasing association with OSA. To assess these results, we consider the MR literature. Specifically, a study by Hartwig et al. (63) reported a comprehensive simulation study investigating the estimated causal effects from two-sample MR when an exposure and outcome association may have various types of confounders, and when using GWAS summary statistics that were and were not adjusted for an important confounder (63), such as BMI. This manuscript indeed had settings that resulted in patterns that we see in the T2D-OSA univariate MR analysis, i.e. where T2D has a protective estimated effect on OSA in BMI-unadjusted analysis, and risk increasing in BMI-adjusted analysis. Based on the simulations in Hartwig et al, we think that a causal structure that may result in such change in direction of the estimated causal effect in univariate MR analysis, is having genetic variants that are common causes of both T2D and BMI (independently, not with BMI mediating their effect on T2D), and in addition having an existing unmeasured common cause of BMI and OSA (see Figure 4 in Hartwig et al.). This is a complicated causal structure, and it is difficult to more specifically hypothesize what the unmeasured confounder that is a common cause of both BMI and OSA. A reasonable hypothesis may be that variants that influence specific fat depots, e.g. visceral or tongue adipose tissue, could be common causes of both traits given recent results showing associations of OSA PRS with adipose tissue distribution measures (64). Future work will need to study MVMR estimation performance in this causal structure (a recent paper applying MVMR to resolve biases caused by covariate adjustment in GWAS did not consider this causal structure (65)), and perhaps use causal structural equation models to (genomic SEMs) to further study the causal structure behind these phenotypes (66)”.

We also edited the methods and the results to report these analyses.

In the methods section (lines 358-372):

“Using BMI-unadjusted summary statistics from published T2D and OSA GWASs described above and also GWAS of BMI (52) we conducted bidirectional multivariable

MR (MVMR) analyses using the TwoSampleMR R package version 0.5.11 to estimate the causal relationship between T2D and OSA, adjusting for BMI as an additional exposure. For each of the two exposures and the outcome variables we selected a set of IVs from the exposure GWASs by: 1) taking the intersection of SNPs between the exposures and outcome GWASs; 2) filtering the resulting SNPs by their p-value in the two exposure GWASs; 3) taking the union of SNPs between the two exposures GWASs; 4) performing clumping of the SNPs that passed the p-value threshold as described above and harmonizing the two exposures and the outcome data for these SNPs to ensure that effect estimates corresponded to the same effect allele across all traits. We performed MVMR analysis based on the harmonized data. From these analyses, we report the estimated causal associations of OSA on T2D adjusted for BMI, and of T2D on OSA, adjusted for BMI. We do not report the estimates of BMI because it is wrong to estimate the effect of BMI on OSA adjusting to T2D (a collider), and similarly for BMI on T2D adjusting for OSA.”

In the results section (lines 545-550):

“We also used MVMR in an attempt to estimate the causal effects of OSA and of T2D while adjusting for BMI as a second exposure. Thus, MVMR used a BMI-unadjusted GWASs. The analysis suggests a modest protective causal effect of T2D on OSA (OR = 0.97, 95% CI [0.95; 0.98]), similar to the BMI-unadjusted MR analysis of T2D causal effect on OSA. In the reverse direction, OSA showed a strong causal effect on T2D (OR = 2.01, 95% CI [1.85; 2.18]). The results are provided in the Supplementary Tables 8.”

With respect to the mediation effect, this is quite small, even in the most extreme case (going from Q0 to Q4, as depicted in the results/figures). Quite honestly, I am not sure how to interpret this mediation; a PRS combines effects from a large number of variants, potentially of pleiotropic effects (especially in the context of T2DM; including the ones involved in energy metabolism, adiposity traits, possibly causing OSA through different pathways). Meaning, genetic risk, as quantified by a PRS for T2DM, will never be not associated with related risk factors. I am not sure what this adds mechanistically or clinically to our understanding of the development of T2DM. I would value MR analyses much more highly than these PRS analyses. Authors should justify the reasoning and implications of PRS analyses performed in this study.

Response: Thank you for this comment. We first would like to clarify that the main goals of this study were to develop a T2DM-PRS that is better suited for individuals of admixed ancestry, and to assess whether poor sleep phenotypes interact with genetic risk to T2DM in that T2DM-PRS have stronger associations with T2DM in individuals with poor sleep phenotypes. Notably,

similar hypotheses have previously addressed other risk factors such as physical activity. Thus, we think that there is a good reason to perform an analysis with PRS.

Regarding the mediation analysis. You are making a good point. Because OSA is a heritable risk factor for T2D, we do expect some mediation: if a genetic variant is causal for OSA, and OSA causes T2D, then the genetic variant is causal for T2D and will be associated with T2D in an analyses that was not adjusted for OSA. Therefore, the PRS should be also associated with OSA, resulting in a mediation effect. Now the questions are (1) is it useful to estimate the proportion of mediation? And (2) is it useful to frame the relationship between OSA and T2D via the mediation lens? We think these are useful.

To clarify that the proportion mediation is fairly modest, we added a precise statement in the conclusion of the abstract:

“These results support a causal association between OSA and DM, with OSA mediating up to 4.7% of the genetic risk for DM.”

We also wrote in the first paragraph of the discussion (lines 722-723):

“Subsequent analysis suggested that a small proportion of the T2D-PRS effect on DM is mediated via an increase in OSA severity.”

...and we also address the question “Is OSA a risk factor of T2D when accounting for BMI, the shared risk factor for both OSA and T2D?”, as discussed in other places in the review.

Additional comments:

- Please double check the affiliations for any typos (e.g. Line 19)

Response: Thank you for pointing this out and we fixed the typo.

- Line 56: Do authors mean T2D, instead of T2D-PRS, given these refer to the MR analysis results?

Response: Thank you and yes, you are correct, we meant to write the T2D here and we fixed the error now.

- Line 59: Conclusion on abstract, in its present form, does not capture the results effectively I think. Within the limitation of character constraints, should include that fact that there is a most mediation of polygenic risk and evidence of a potentially bi-directionally causal link between T2DM and OSA.

Response: Thank you for this helpful suggestion. We agree that the original conclusion could better highlight the findings of our work. We have revised the abstract conclusion to highlight both the substantial mediation of T2D-PRS by OSA and the potential bidirectional causal relationship between T2DM and OSA:

“These results support a causal association between OSA and DM, with OSA mediating up to 4.7% of the genetic risk for DM.”

- Lines 80-82: It would be helpful to briefly expand on these potential pathways given it is not readily apparent to the reader

Response: Thank you for this helpful suggestion. We reorganized the introduction to improve the flow based on your suggestions. Now, in the first paragraph of the introduction we have information about potential pathways that was previously in a different paragraph (new text in bold):

“OSA is a common sleep-related breathing disorder characterized by repeated episodes of upper airway obstruction associated with intermittent hypoxemia and fragmented sleep- mechanisms that are implicated in impaired glucose regulation (2). The potential pathways linking OSA and DM and evidence for a causal association have been reported previously (3–6). **For example, it was shown that OSA-related intermittent hypoxia and fragmented sleep increase the risk of DM via effects on chronic inflammation, oxidative stress and metabolic imbalance that adversely affect pancreatic function, hormonal regulation, and insulin resistance (7–10).**”

- Lines 95-97: Authors indicate that a prior analysis in HCHS/SOL using 2-sample MR showed DM and glycemic traits exhibit a causal association with OSA but not the other way around. Then authors, in the following sentence, go on to say that these results indicate the existence of a potential bidirectional association between OSA and diabetes. This is seemingly clarified between the lines 99-104, however rewording can avoid the confusion further.

Response: We apologize for these tangled arguments. Indeed, the previous paragraph already introduced one causal direction (OSA to diabetes). We reworded this paragraph as follows:

“Recent analysis in HCHS/SOL utilized genetic techniques to study the association of OSA with a range of phenotypes. The study showed that a polygenic risk score for OSA was associated with glycemic traits (19). Additionally, the same study applied two-sample Mendelian randomization (MR) analysis using GWAS summary statistics, and suggested a causal effect of DM-related glycemic traits on OSA. Indeed, several large cohort studies have reported observational evidence of a bidirectional association between OSA and diabetes (16). However, this MR analysis found no statistically significant association of OSA on DM-related traits. This null finding suffers from limitations: the MR used OSA GWAS summary statistics from a population of Finnish Europeans, and included only a small number of genetic loci. In addition, the association between OSA and diabetes risk may involve more complex mechanisms such as effect modification. For example, OSA may interact with genetic risk factors, affecting the likelihood of developing diabetes.”

- Lines 221-224: What reference panel was used to infer these four ancestral populations? If it is 1000 genomes, AMR would stand for ‘admixed american’ and these include Mexicans and Puerto Ricans from the US, Colombians, and Peruvians (thus, although including an admixed Hispanic/Latino population, not really Native American [Amerindian] i.e., a homogenous continental ancestry group).

Response: You are correct. We used 1000 Genome (we checked and it is written in the methods). We now highlight this as a limitation in the discussion paragraph that focuses on the PRSs:

“Finally, another limitation of gapPRSsum is that 1000 Genome AMR reference panel does not perfectly match the Amerindian ancestry proportions estimated in HCHS/SOL, as the latter correspond to homogeneous Amerindian (Native American ancestry), while the 1000 Genome AMR reference population includes admixed American individuals.”

- Lines 227-230: Were strand ambiguous variants removed prior to PRS construction? Please clarify.

Response: Thank you for the comment and we confirm that the strand-ambiguous variants were removed prior to PRS construction. We agree, the text needs more clarification. We revised the sentence to:

“Only variants with imputation quality $R^2 \geq 0.8$, minor allele frequency ≥ 0.01 , and not strand-ambiguous, were used in PRS construction.”

- Lines 232-233: *Were PRS standardized to mean 0 and variance 1 prior to each analyses? If not, regression betas will not exactly correspond to increase in trait/odds per 1 SD increase in PRS. Please clarify and revise as needed.*

Response: No, we did not standardize them separately for each analysis. We standardized them over the entire dataset once, then used the same values in consequent analysis. We updated the methods section (new text in bold):

“After summing the ancestry-specific PRS, we again standardized each resulting PRS measure. These standardized PRSs were subsequently used and were not transformed again.”

- Lines 257-258: *This is an unusual way of obtaining CIs for AUC. Why did authors not use bootstrapping as commonly done in the field? Please justify.*

Response: We chose to use repeated random train-test splits (90% training, 10% testing) over 500 iterations to estimate the variability of AUC on an independent test dataset, rather than on the training dataset (bootstrap would estimate the AUC over a range of realistic dataset, but in each case the AUC would be estimated on a training dataset, which often over-estimates prediction performance). We computed the 95% confidence interval using the empirical 2.5 and 97.5 percentiles of the AUC distribution. We have updated the text to clarify this point.

“We computed the area under the receiver operating characteristic curve (AUC) and its 95% confidence interval (CI) for each model using repeated random train-test splits, to estimate model performance on an independent dataset (rather than on a training dataset, which may lead to optimism due to overfitting). Specifically, we randomly split the data into training (90%) and testing (10%) sets, trained the model on the training data, and evaluated its performance on the testing data using the *predict* function in R. This process was repeated 500 times, and the AUC was computed using the *auc* function from the Metrics R library (version 0.1.4). The mean AUC was reported, and the 95% CI was defined using the 2.5 and 97.5 percentiles of the 500 AUC values.”

- Lines 310-312: *This sentence is written a bit informally, especially the latter part, please consider revising.*

Response: Thank you for this suggestion, we agree, and revised the sentence to:

“Thus, we estimated the proportion of the T2D-PRS effect on DM risk that is mediated through increased risk of mild-to-severe OSA, comparing individuals whose T2D-PRS values correspond to, as an example, the 0.25 quantile (control value) versus the 0.75 quantile (treatment value) of the PRS distribution.”

- *Line 461: This subsection title is a very bold claim. MR analyses, although robust to some of the challenges of observational studies, are not infallible given its strong assumptions (some of which are not verifiable despite all sensitivity analyses and other advances in methodology). Please temper the language to reflect that (such as ‘... evidence that T2D potentially has a causal association with OSA’ or similar). I gave line 461 as an example, however, all MR analysis results discussed in the text should reflect this.*

Response: Thank you for this suggestion. We have revised this subsection title and also text related to MR results to more cautiously reflect the strength of the evidence.

Specifically, we edited the subsection title as follows:

- “Genetic analysis suggests a potential causal effect of OSA on T2D”

Additionally, we reviewed the language in other places and adapted it accordingly.

- *Lines 501-504: Authors should also include here that the size of this effect, in addition to not being significant, was also quite modest, in order to avoid unintentionally misleading the readership.*

Response: Thank you for this helpful suggestion. We revised the sentence to say:

“T2D-PRSs were highly associated with DM, and, contrary to our expectation, their association with DM appeared weaker in individuals with moderate-to-severe OSA (estimated OR=2.16) compared to individuals with no (OR=2.81) or mild (OR=3.14) OSA. However, the interaction test was not statistically significant.”

- *Lines 504-505: Again, quite modest percentage of mediation, please be clear to communicate this.*

Response: Thank you for this helpful suggestion. We revised the sentence which now says:

“Subsequent analysis suggested that a small proportion of the T2D-PRS effect on DM is mediated via an increase in OSA severity.”

- Figure 1 color legends for panels b,c, and d are not clear. I am honestly not sure what I am looking at in terms of groups for ORs. Please revise the figure to make it very clear what OR corresponds to which analysis/group.

Response: Thank you for this comment. You are absolutely right - the legends for panels b, c, and d were missing due to overlapping elements caused by space constraints in the original layout. We have revised Figure 1 to clearly display all legends, so that the ORs and corresponding groups/analyses are now easy to interpret.

- Figure 2: Please increase the point size for the dots denoting the OR value. The way it is right now, it almost looks like just a horizontal line depicting the CIs.

Response: Thank you for your comment, we agree and we revised the Figure 2 so the OR points are more prominent.

Reviewer #2 (Remarks to the Author):

This study systematically analyzes how certain components of the polygenic risk score (PRS) for type 2 diabetes mellitus (T2DM) may mediate the relationship between obstructive sleep apnea (OSA) and the incidence of T2DM. The background, methodology, and results are well-explained and clearly articulated throughout the manuscript. Additionally, the authors' approach of addressing the challenge of a mixed population by comparing and applying various PRS methodologies is commendable and demonstrates a thoughtful strategy. Below, I have provided comments on areas that could benefit from further refinement:

Response: Thank you for your review!

Comments. : Figures 1 and 2 - Bar Chart Formatting: The bar lines in Figures 1 and 2 could be slightly thicker to enhance clarity and visual distinction between data points. Additionally, the legends appear too small, making them difficult to read. Adjusting the font size of the legends to improve readability would greatly aid in understanding the charts.

Response: Thank you for your comment. We have revised the Figure 1 and 2 and made the bar lines thicker and increased the font size of the legends.

Reviewer #3 (Remarks to the Author):

This manuscript investigates the association between diabetes and obstructive sleep apnea in the Hispanic/Latino population, utilizing multi-ancestry type 2 diabetes polygenic risk scores (T2D-PRSs). The findings are of interest and the study addresses an important question; however, there is room for improvement in the clarity and presentation of the results, particularly regarding the figures and associated descriptions.

Response: Thank you for your review!

Comments:

1. In Introduction section, the sentence "it is possible that there is a more complex association between OSA and risk of diabetes" is somewhat ambiguous, as it is not clear what the comparison is being made against. Clarifying what is meant by "more complex" could improve the reader's understanding.

Response: Thank you and agreed. We revised this sentence and added clarification as follows:

"In addition, the association between OSA and diabetes risk may involve more complex mechanisms such as effect modification. For example, OSA may interact with genetic risk factors, affecting the likelihood of developing diabetes."

2. The final sentence of the second paragraph in Introduction, ("OSA may modify genetic risk of diabetes") feels somewhat abrupt and underdeveloped. Since the following paragraph discusses genetic risk in more detail, this sentence may be unnecessary or better integrated with the next section to maintain a more coherent flow.

Response: Thank you for this comment. We agree with your suggestion and have revised the relevant text to improve clarity and flow. Specifically, we reworded and repositioned the sentence to better integrate with the following discussion of genetic risk (as described in our response to your *Comment 1*).

3. *In the second sentence in the third paragraph of the Introduction, it may be helpful to clarify that intermittent hypoxia and fragmented sleep are characteristic consequences of OSA.*

Response: Thank you for this helpful suggestion. We revised the sentence and re-arranged the text and improve the overall clarity:

“For example, it was shown that OSA-related intermittent hypoxia and fragmented sleep increase the risk of DM via effects on chronic inflammation, oxidative stress and metabolic imbalance that adversely affect pancreatic function, hormonal regulation, and insulin resistance (7–10).”

4. *In the third paragraph of the Introduction, the authors refer to genetic studies showing interactions between sleep duration and genetic loci in relation to blood pressure and lipid levels, suggesting that similar mechanisms might underlie associations between OSA and diabetes. To strengthen the rationale, the authors might consider noting whether previous genetic studies have reported interactions between sleep-related traits and glycemic outcomes or diabetes risk. If such evidence exists, its inclusion would further support the study’s premise.*

Response: This is a great suggestion. Unfortunately, while there is an ongoing work from the CHARGE consortium (looking at short and long sleep interactions with genetic effects on diabetes), it is advancing slowly and still far from being published.

5. *In the Methods section, diabetes mellitus (DM) is defined according to the American Diabetes Association (ADA) criteria. In addition, DM status was also ascertained using scanned medication data at baseline and self-reported medication use at the second exam. It would be helpful to clarify what specific medications were considered indicative of DM in this context. Is there a possibility that some individuals who did not meet the ADA criteria may have been using diabetes medications for other purposes, such as early intervention, treatment of other conditions, or off-label use (e.g., for weight loss)? Furthermore, it would strengthen the rigor of the study to report how many participants were classified as having DM based on ADA criteria versus medication data. In addition, how many participants underwent blood tests, and what were the reasons for missing data, if any, in participants who did not? Also, the sentence stating that “ADA criteria rely on laboratory tests, either [...], or scanned/transcribed anti-diabetic medication use” may be misleading. The use of anti-diabetic medications is not formally included in the ADA diagnostic criteria.*

Response: Thank you for this useful comment. First, we deleted this unnecessary and misleading sentence part “or scanned/transcribed...”. Now the definition that we used is accurately reported.

Second, we added the following clarification in the Supplementary Note 2:

Scanned anti-diabetic medications included: “antidiabetics” , “insulin” , “mixed insulin” , “beef insulin” , “pork insulin” , “human insulin” , “antidiabetic - amylin analogs” , “sulfonylureas” , “sulfoylurea combinations” , “antidiabetic - amino acid derivatives” , “antidiabetic-d-phenylalanine derivatives” , “biguanides” , “meglitinide analogues” , “meglitinide analogues” , “diabetic other” , “diabetic other - combinations” , “aldose reductase inhibitors” , “alpha-glucosidase inhibitors” , “insulin sensitizing agents” , “thiazolidinediones” , “antidiabetic combinations” , “sulfonylurea-biguanide combinations” . Self-reported antidiabetic medication use was determined by the “Yes” answer to following question: “Were any of the medications you took during the last 4 weeks for high blood sugar or diabetes?” .

Third, we report the number of participants that were classified as having DM based on ADA criteria versus medication data as follows:

“There were only 17 people with anti-diabetes classified based on medication data, who were not classified as diabetics by ADA guidelines.”

We also added this statement, referring to potential missing values and diabetes categorization:

“Diabetes status was considered missing if fasting glucose, post-OGTT, HbA1c, and information on anti-diabetic medications were all unavailable. Participants without glucose lab data and no record of anti-diabetic medication use were assumed to have normal glucose regulation.”

Unfortunately, we cannot postulate reasons for individuals not having lab values.

6. In the Results section, the authors refer to “hyperglycemic” participants; however, no definition of hyperglycemia is provided in the Methods section. To improve clarity and reproducibility, it is recommended that the criteria used to define hyperglycemia be explicitly stated in the Methods.

Response: Thank you, we agree, and added the definition for clarity:

“Hyperglycemia (pre-diabetic status) was defined by the ADA guidelines based on laboratory tests of fasting time > 8 and fasting glucose in range 100 to 125 mg/dL, or post-OGTT glucose in range 140 to 199 mg/dL, or $5.7\% \leq A1C < 6.5\%$.”

7. To enhance the interpretability and standalone value of the figures (including figures in supplementary material), it is recommended that the reference group or scale for both the ORs and IRRs be clearly specified in each figure and its corresponding caption. Currently, it is not immediately clear from the figures what these estimates reflect.

Response: This is now fixed. For example, we clarify that estimates are per 1 SD increase in the PRS; we clarify that, in PRS associations with poor sleep health phenotypes the compared groups are individuals without the specific poor sleep phenotype, and we clarified that percent mediation (in Figure 2) was estimated for specific values of the PRS, selected as distribution quantiles. We changed figure captions in both the main and supplementary text.

8. In Figure 1, panels b and c do not include legends identifying the plotted groups or lines, which makes interpretation difficult. In panel d, although a legend is provided, each group appears to have two separate plotted values. Based on the caption, it can be inferred that these represent DM and incident DM, but it is not clearly labeled which is which. To improve clarity and allow for independent interpretation of the figure, it is recommended that all panels include appropriate legends and that the meaning of the plotted values be clearly indicated, particularly in panel d.

Response: Thank you for your comment, and you are right - the legends for Figure 1 panels b and c were missing due to overlapping elements. We have revised the figure to include missing legends. In panel d we have increased the font size for each of the results groups: “Cross-sectional analysis” and “Incident analysis” and also increased the font size of the legends.

9. The figure captions, use the phrasing "OR of the T2D-PRSs" and "IRR of the T2D-PRSs," which may be misleading, as they suggest the PRS is the outcome. To avoid confusion, it would be clearer to rephrase as "OR for DM" or "IRR for incident DM" to reflect that DM is the outcome and PRS is the predictor variable.

Response: Agreed, this is now fixed.

10. In Figure 2a, the addition of a label or legend would improve interpretability, as it is not

immediately clear what the plotted data represent. Additionally, in panels c and d, although labels are present, each group appears to be plotted twice, and the distinction is not clearly labeled in the figure itself. To improve clarity, it is recommended that all plotted elements be explicitly identified, so that the figure can be understood independently of the main text.

Response: Thank you for your comment, we agree and we improved the clarity of the figure to clearly display the categories in with the association was estimated and also added the description to the Figure's 2a x-axis to explicitly say that the results are shown for mgbPRSsum, in association with poor sleep phenotypes. We also clarified panels c and d in the panel titles and in the legend.

Reviewer #4 (Remarks to the Author):

Dear Editor

I am grateful for the invitation and opportunity to review the manuscript "COMMSMED-24-1357-T Obstructive sleep apnea mediates genetic risk of Diabetes Mellitus: The Hispanic Community Health Study/Study of Latinos". The manuscript assessed the genetic risk between obstructive sleep apnea and diabetes.

In general, the manuscript presents consistent content and structure. Reading, vocabulary and interpretation are easy to understand. The methodology seems appropriate for the proposed objectives and the results are compatible with the proposed objectives. In addition, the discussion presents classic and current studies, thus generating new ideas for future studies.

Response: Thank you for your review and helpful suggestions!

I begin this review by contextualizing the role of variables related to lifestyle in the regression model to assess the strength of association. Dietary habits such as the absence of industrialized and ultra-processed foods, sleep patterns and the sleep-wake cycle no longer modulated by the "moon-sun" cycle but recently modulated by electrical energy, pedestrian commuting habits, among others, very common in the past, were very different from the current lifestyle. Thus, exploring the influence of these variables on the magnitude of the strength of association between OSA and diabetes may be timely and provide insights for understanding complex associations such as OSA and metabolic profile. Based on this comment, I suggest: Consider other measures of obesity in the regression model

Response: Thank you, we agree that it is useful to provide these comparisons as well. We just want to clarify that we did not directly consider OSA and diabetes association (other than in the mediation analysis), rather, association analyses relied on genetic analysis techniques. Below, the added analyses therefore estimate the associations of T2D PRS with OSA.

1.1 The authors used BMI as an obesity metric. Considering the role of visceral fat in mediating the results. Would replacing BMI with a measure of central obesity such as abdominal circumference or waist-to-hip ratio potentially provide more robust evidence on the strength of association in the model?

2 - Potentially confounding variables:

2.1- Medications: Did the participants regularly use medications known for their anti-inflammatory effects such as statins?

2.2- Socioeconomic status

2.3 - Physical activity: an important aspect to be included in the model, since from an evolutionary point of view, humans actively allocated their energy expenditure through walking, unlike the current sedentary behavior

3- Race – ancestry in the genetic model: Could one of the limitations of the model be the fact that Latinos also have indigenous ancestry?

Response:

We thank the reviewer for all these valuable suggestions and we addressed these by:

1.1 performing a secondary analysis of association between DM PRSs and OSA outcomes using waist-to-hip ratio (WHR) as the obesity measure instead of BMI as a covariate.

2 We estimated the association of DM PRSs with OSA outcomes adjusting for the following additional confounders:

2.1 Medications: statins

2.2 Socioeconomic status: education level, range of income

2.3 Total physical activity, vigorous physical activity

2.4 Hispanic/Latino Background

The results showed that when adjusting for WHR the association results were similar to the results when adjusting for BMI. We added Supplementary Figure 12 which shows the comparison of the results between the two variables. When adjusting for WHR the association was not statistically significant between T2D-PRSs and moderate-to-severe OSA, however, when adjusting for BMI, all three T2D-PRSs had statistically significant association with both mild-to-severe OSA and moderate-to-severe OSA.

When adjusting for confounders listed above, the estimated association between DM PRSs and OSA (both mild-to-severe OSA and moderate-to-severe OSA) was statistically significant for all three types of DM PRS (other than two associations with gapPRSsum for which the 95% overlaps the null value) and the effect estimates barely change. We updated the manuscript to reflect these results (lines 737-747):

“Adjusting for WHR instead of BMI resulted in a statistically significant association between all T2D-PRSs and mild-to-severe OSA (ORs = 1.1 – 1.17). Complete results are provided in the Supplementary Table 6. Comparison of the results between BMI and WHR are visualized in the Supplementary Figure 12. In another analysis, when adjusting for the additional potential confounders (medication, socioeconomic status, physical activity and self-reported Hispanic/Latino background) the association between all T2D-PRSs and both mild-to-severe OSA and moderate-to-severe OSA remained about the same across analyses (Supplementary Table 7).”

STRENGTHS OF THE STUDY:

1 - the evaluation of the association between OSA and diabetes from the point of view of pleiotropy brings new information to the understanding of the complex metabolic imbalance of this mechanistic cascade of pathophysiological events. This approach is relatively recent and additive to information from cross-sectional studies.

2 – The epidemiological characteristic, the robustness of the association measures.

No further comments

There were no comments to address (only the editorial check list, provided in a different file).